# ReconX: Reconstruct Any Scene from Sparse Views with Video Diffusion Model

## Abstract

Advancements in 3D scene reconstruction have transformed 2D images from the real world into 3D models, producing realistic 3D results from hundreds of input photos. Despite great success in dense-view reconstruction scenarios, rendering a detailed scene from sparse views is still an ill-posed optimization problem, often resulting in artifacts and distortions in unseen areas. In this paper, we propose **ReconX**, a novel 3D scene reconstruction paradigm that reframes the ambiguous reconstruction problem as a temporal generation task. The key insight is to unleash the strong generative prior of large pre-trained video diffusion models for sparse-view reconstruction. Nevertheless, it is challenging to preserve 3D view consistency when directly generating video frames from pre-trained models. To address this issue, given limited input views, the proposed ReconX first constructs a global point cloud and encodes it into a contextual space as the 3D structure condition. Guided by the condition, the video diffusion model then synthesizes video frames that are detail-preserved and exhibit a high degree of 3D consistency, ensuring the coherence of the scene from various perspectives. Finally, we recover the 3D scene from the generated video through a confidence-aware 3D Gaussian Splatting optimization scheme. Extensive experiments on various real-world datasets show the superiority of ReconX over state-of-the-art methods in terms of quality and generalizability.

## 1 Introduction

With the rapid development of photogrammetry techniques such as NeRF (Mildenhall et al., 2020) and 3D Gaussian Splatting (3DGS) (Kerbl et al., 2023), 3D reconstruction has become a popular research topic in recent years, finding various applications from virtual reality (Dalal et al., 2024) to autonomous navigation (Adamkiewicz et al., 2022) and beyond (Martin-Brualla et al., 2021b; Liu et al., 2024a; Wu et al., 2024a; Charatan et al., 2024). However, sparse-view reconstruction is an ill-posed problem (Gao et al., 2024; Yu et al., 2021) since it involves recovering a complex 3D structure from limited viewpoint information (*i.e.,* even as few as two images) that may correspond to multiple solutions. This uncertain process requires additional assumptions and constraints to yield a viable solution.

Recently, powered by the efficient and expressive 3DGS (Kerbl et al., 2023) with fast rendering speed and high quality, several feed-forward Gaussian Splatting methods (Charatan et al., 2024; Szymanowicz et al., 2024b; Chen et al., 2024a) have been proposed to explore 3D scene reconstruction from sparse view images. Although they can achieve promising interpolation results by learning scene-prior knowledge from feature extraction modules (*e.g.,* epipolar transformer (Charatan et al., 2024)), insufficient captures of the scene still lead to an ill-posed optimization problem (Wu et al., 2024b). As a result, they often suffer from severe artifact and implausible imagery issues when rendering the 3D scene from novel viewpoints, especially in unseen areas.

To address the limitations, we propose **ReconX**, a novel 3D scene reconstruction paradigm that reformulates the inherently ambiguous reconstruction problem as a generation problem. Our key insight is to unleash the strong generative prior of pre-trained large video diffusion models (Blattmann et al., 2023a;b; Xing et al., 2023) to create more observations for the downstream reconstruction task. Despite the capability to synthesize video clips featuring plausible 3D structures (Gao et al., 2024), recovering a high-quality 3D scene from current video diffusion models is still challenging,

Figure 1: **An overview of our ReconX framework for sparse-view reconstruction.** Unleashing the strong generative prior of video diffusion models, we can create more observations for 3D reconstruction and achieve impressive performance.

due to the poor 3D view consistency across generated 2D frames. Grounded by theoretical analysis, we explore the potential of incorporating 3D structure condition into the video generative process, which bridges the gap between the under-determined 3D creation problem and the fully-observed 3D reconstruction setting. Specifically, given sparse images, we first build a global point cloud through a pose-free stereo reconstruction method. Then we encode it into a rich context representation space as the 3D condition in cross-attention layers, which guides the video diffusion model to synthesize detail-preserved frames with 3D consistent novel observations of the scene. Finally, we reconstruct the 3D scene from the generated video through Gaussian Splatting with a 3D confidence-aware and robust scene optimization scheme, which further deblurs the uncertainty in video frames effectively. Extensive experiments verify the efficacy of our framework and show that ReconX outperforms existing methods for high quality and generalizability, revealing the great potential to craft intricate 3D worlds from video diffusion models. The overview and examples of reconstructions are shown in Figure 1.

In summary, our main contributions are as follows:

- We introduce ReconX, a novel sparse-view 3D scene reconstruction framework that reframes the ambiguous reconstruction challenge as a temporal generation task.
- We incorporate the 3D structure condition into the conditional space of the video diffusion model to generate 3D consistent frames and propose a 3D confidence-aware optimization scheme in 3DGS to reconstruct the scene given the generated video.
- Extensive experiments demonstrate that our ReconX outperforms existing methods for high-fidelity and generalizability on a variety of real-world datasets.

## 2 RELATED WORK

**Sparse-view reconstruction.** NeRF and 3DGS typically demand hundreds of input images and rely on the multi-view stereo reconstruction (MVS) approach (*e.g.,* COLMAP (Schönberger & Frahm, 2016)) to estimate the camera parameters. To address the issue of low-quality 3D reconstruction caused by sparse views, PixelNeRF (Yu et al., 2021) proposes using convolutional neural networks to extract features from the input context. Moreover, FreeNeRF (Yang et al., 2023) adopts the frequency and density regularized strategies to alleviate the artifacts caused by insufficient inputs without any additional cost. To mitigate the overfitting to input sparse views in 3DGS, FSGS (Zhu et al., 2023) and SparseGS (Xiong et al., 2023) employ a depth estimator to regularize the optimization process. However, these methods all require known camera intrinsics and extrinsics, which is not practical in real-world scenario. Benefiting from the existing powerful 3D reconstruction model (*i.e.,* DUSt3R (Wang et al., 2024a)), InstantSplat (Fan et al., 2024) is able to acquire accurate camera parameters and initial 3D representations from unposed sparse-view inputs, leading to the efficient and high-quality 3D reconstruction.

**Regression model for generalizable view synthesis.** While NeRF and 3DGS are optimized per-scene, a line of research aims to train feed-forward models that output a 3D representation directly

from a few input images, bypassing the need for time-consuming optimization. Splatter image (Szymanowicz et al., 2024b) performs an efficient feed-forward manner for monocular 3D object reconstruction by predicting a 3D Gaussian for each image pixel. Meanwhile, pixelSplat (Charatan et al., 2024) proposes predicting the scene-level 3DGS from the image pairs, using the epipolar transformer to better extract scene features. Following that, MVsplat (Chen et al., 2024a) introduces the cost volume and depth refinements to produce a clean and high-quality 3D Gaussians in a faster way. LatentSplat (Wewer et al., 2024) encodes the variational 3D Gaussians and utilizes a discriminator to synthesize more realistic images. To reconstruct a complete scene from a single image, Flash3D (Szymanowicz et al., 2024a) adopts a hierarchical 3DGS learning policy and depth constraint to achieve high-quality interpolation and extrapolation view synthesis. Although these methods leverage the 3D data priors, they are limited by the scarcity and diversity of 3D data. Consequently, these methods struggle to achieve high-quality renderings in unseen areas, especially when out-of-distribution (OOD) data is used as input.

**Generative models for 3D reconstruction.** Constructing comprehensive 3D scenes from limited observations demands generating 3D content, particularly for unseen areas. Earlier studies distill the knowledge in the pre-trained text-to-image diffusion models (Rombach et al., 2022; Saharia et al., 2022; Ramesh et al., 2022) into a coherent 3D model. Specifically, the Score Distillation Sampling (SDS) technique (Wu et al., 2024b; Lin et al., 2023; Liu et al., 2024c; Wang et al., 2024b) is adopted to synthesize a 3D object from the text prompt. To enhance the 3D consistency, several approaches (Wu et al., 2024a; Shi et al., 2023; Liu et al., 2023) inject the camera information into diffusion models, providing strong multi-view priors. Furthermore, ZeroNVS (Sargent et al., 2023) and CAT3D (Gao et al., 2024) extend the multi-view diffusion to the scene level generation. GeNVS (Chan et al., 2023) embeds a 3D feature field into the diffusion model to enhance the novel view synthesis ability. More recently, video diffusion models (Blattmann et al., 2023a; Xing et al., 2023) have shown an impressive ability to produce realistic videos and are believed to implicitly understand 3D structures (Liu et al., 2024b). SV3D (Voleti et al., 2024) and V3D (Chen et al., 2024b) explore fine-tuning the pre-trained video diffusion model for 3D object generation. Meanwhile, MotionCtrl (Wang et al., 2024c) and CameraCtrl (He et al., 2024) achieve scene-level controllable video generation from a single image by explicitly injecting the camera pose into video diffusion models. However, they cannot work for the unconstrained sparse-view 3D scene reconstruction, which requires strong 3D consistency.

## 3 MOTIVATION FOR RECONX

In this paper, we focus on the fundamental problem of 3D scene reconstruction and novel view synthesis (NVS) from very sparse view (*e.g.,* as few as two) images. Most existing works (Chen et al., 2024a; Yu et al., 2021; Charatan et al., 2024; Szymanowicz et al., 2024a) utilize 3D prior and geometric constraints (*e.g.,* depth, normal, cost volume) to fill the gap between observed and novel regions in sparse-view 3D reconstruction. Although capable of producing highly realistic images from the given viewpoints, these methods often struggle to generate high-quality images in areas not visible from the input perspectives due to the inherent problem of insufficient viewpoints and the resulting instability in the reconstruction process. To address this issue, a natural idea is to create more observations to convert the under-determined 3D creation problem into a fully constrained 3D reconstruction setting. Recently, video generative models have shown promise for synthesizing video clips featuring 3D structures (Voleti et al., 2024; Blattmann et al., 2023a; Xing et al., 2023). This inspires us to unleash the strong generative prior of large pre-trained video diffusion models to create temporal consistent video frames for sparse-view reconstruction. Nevertheless, it is non-trivial as the main challenge lies in poor 3D view consistency among video frames, which significantly limits the downstream 3DGS training process. To achieve 3D consistency within video generation, we first analyze the video diffusion modeling from a 3D distributional view. Let $\boldsymbol{x}$ be the set of rendering 2D images from any 3D scene in the world, $q(\boldsymbol{x})$ be the distribution of the rendering data $\boldsymbol{x}$, and our goal is to minimize the divergence $\mathcal{D}$:

$$\min_{\boldsymbol{\theta} \in \Theta, \psi \in \Psi} \mathcal{D}\left(q(\boldsymbol{x}) \| p_{\boldsymbol{\theta}, \psi}(\boldsymbol{x})\right), \tag{1}$$

where $p_{\boldsymbol{\theta}, \psi}$ is a diffusion model parameterized by $\boldsymbol{\theta} \in \Theta$ (the parameters in the backbone) and $\psi \in \Psi$ (any embedding function shared by all data). The vanilla video diffusion model (Xing et al., 2023) chooses a CLIP (Radford et al., 2021) model $g$ to add an image-based condition (*i.e.,*

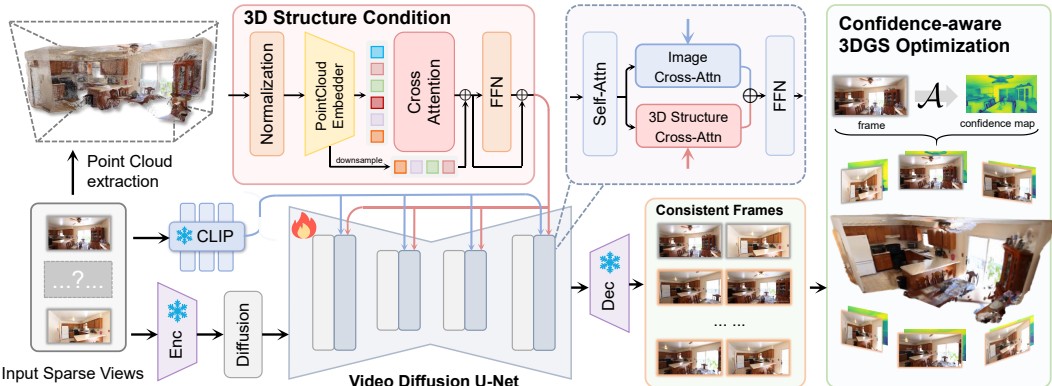

Figure 2: **Pipeline of ReconX.** Given sparse-view images as input, we first build a global point cloud and project it into 3D context representation space as 3D structure condition. Then we inject the 3D structure condition into the video diffusion process and guide it to generate 3D consistent video frames. Finally, we reconstruct the 3D scene from the generated video through Gaussian Splatting with a 3D confidence-aware and robust scene optimization scheme. In this way, we unleash the strong power of the video diffusion model to reconstruct intricate 3D scenes from very sparse views.

$\psi = g$). However, in sparse-view 3D reconstruction, only conditioning on 2D images cannot provide sufficient condition for approximating $q(\boldsymbol{x})$ (Charatan et al., 2024; Chen et al., 2024a; Wu et al., 2024b). Motivated by this, we explore the potential of incorporating the native 3D prior (denoted by $\mathcal{F}$) to find an optimal solution in Equation 1 and derive a theoretical formulation for our analysis in Proposition 1.

**Proposition 1.** *Let $\boldsymbol{\theta}^*, \psi^* = g^*$ be the optimal solution of the solely image-based conditional diffusion scheme and $\tilde{\boldsymbol{\theta}}^*, \tilde{\psi}^* = \{g^*, \mathcal{F}^*\}$ be the optimal solution of the diffusion scheme with a native 3D prior. Suppose the divergence $\mathcal{D}$ is convex and the embedding function space $\Psi$ includes all measurable functions, then we have $\mathcal{D}(q(\boldsymbol{x}) \| p_{\tilde{\boldsymbol{\theta}}^*, \tilde{\psi}^*}(\boldsymbol{x})) < \mathcal{D}(q(\boldsymbol{x}) \| p_{\boldsymbol{\theta}^*, \psi^*}(\boldsymbol{x}))$. (Proof in our Appendix)*

Towards this end, we *reformulate the inherently ambiguous reconstruction problem as a generation problem* by incorporating a 3D native structure condition into the diffusion process. More preliminaries can be found in our Appendix.

## 4 METHOD

### 4.1 OVERVIEW OF RECONX

Given $K$ sparse-view (*i.e.,* as few as two) images $\mathcal{I} = \{\boldsymbol{I}^i\}_{i=1}^{K}$, $(\boldsymbol{I}^i \in \mathbb{R}^{H \times W \times 3})$, our goal is to reconstruct the underlying 3D scene, where we can synthesize novel views of unseen viewpoints. In our framework ReconX, we first build a global point cloud $\mathcal{P} = \{\boldsymbol{p}_i, 1 \le i \le N\} \in \mathbb{R}^{N \times 3}$ from $\mathcal{I}$ and project $\mathcal{P}$ into the 3D context representation space $\mathcal{F}$ as the structure condition $\mathcal{F}(\mathcal{P})$ (Sec. 4.2). Then we inject $\mathcal{F}(\mathcal{P})$ into the video diffusion process to generate 3D consistent video frames $\mathcal{I}' = \{\boldsymbol{I}^i\}_{i=1}^{K'}$, $(K' > K)$, thus creating more observations (Sec. 4.3). To alleviate the negative artifacts caused by the inconsistency among generated videos, we utilize the confidence maps $\mathcal{C} = \{\mathcal{C}_i\}_{i=1}^{K'}$ from the DUSt3R model and LPIPS loss (Zhang et al., 2018a) to achieve a robust 3D reconstruction (Sec. 4.4). In this way, we can unleash the full power of the video diffusion model to reconstruct intricate 3D scenes from very sparse views. Our pipeline is depicted in Figure 2.

### 4.2 BUILDING THE 3D STRUCTURE CONDITION

Grounded by the theoretical analysis in Sec. 3, we leverage an unconstrained stereo 3D reconstruction method DUSt3R (Wang et al., 2024a) with point-based representations to build the 3D structure condition $\mathcal{F}$. Given a set of sparse images $\mathcal{I} = \{\boldsymbol{I}^i\}_{i=1}^{K}$, we first construct a connectivity graph

$\mathcal{G}(\mathcal{V}, \mathcal{E})$ of $K$ input views similar to DUSt3R, where vertices $\mathcal{V}$ and each edge $e = (n, m) \in \mathcal{E}$ indicates that the images $\boldsymbol{I}^n$ and $\boldsymbol{I}^m$ shares visual contents. Then we use $\mathcal{G}$ to recover a globally aligned point cloud $\mathcal{P}$. For each image pair $e = (n, m)$, we predict pairwise pointmaps $P^{n,n}, P^{m,n}$ and their corresponding confidence maps $\mathcal{C}^{n,n}, \mathcal{C}^{m,n} \in \mathbb{R}^{H \times W \times 3}$. For clarity, we denote $P^{n,e} := P^{n,n}$ and $P^{m,e} := \mathcal{P}^{m,\bar{n}}$. Since we aim to rotate all pairwise predictions into a shared coordinate frame, we introduce transformation matrix $T_e$ and scaling factor $\sigma_e$ associated with each pair $e \in \mathcal{E}$ to optimize global point cloud $\mathcal{P}$ as:

$$\mathcal{P}^* = \arg\min_{\mathcal{P}, T, \sigma} \sum_{e \in \mathcal{E}} \sum_{v \in e} \sum_{i=1}^{HW} \mathcal{C}_i^{v,e} \, \|\mathcal{P}_i^v - \sigma_e T_e P_i^{v,e}\|. \tag{2}$$

More details of the point cloud extraction can be found in Wang et al. (2024a). Having aligned the point clouds $\mathcal{P}$, we now project it into a 3D context representation space $\mathcal{F}$ through a transformer-based encoder for better interaction with latent features of the video diffusion model. Specifically, we embed the input point cloud $\mathcal{P}$ into a latent code using a learnable embedding function and a cross-attention encoding module:

$$\mathcal{F}(\mathcal{P}) = \text{FFN} \left( \text{CrossAttn}(\text{PosEmb}(\tilde{\mathcal{P}}), \text{PosEmb}(\mathcal{P})) \right), \tag{3}$$

where $\tilde{\mathcal{P}}$ is a down-sampled version of $\mathcal{P}$ at $1/8$ scale to efficiently distill input points to a compact 3D context space. Finally, we get the 3D structure guidance $\mathcal{F}(\mathcal{P})$ which contains sparse structural information of the 3D scene that can be interpreted by the denoising U-Net.

### 4.3 3D Consistent Video Frames Generation

In this subsection, we incorporate the 3D structure condition $\mathcal{F}(\mathcal{P})$ into the video diffusion process to obtain 3D consistent frames. To achieve consistency between generated frames and high-fidelity rendering views of the scene, we utilize the video interpolation capability to recover more unseen observations, where the first frame and the last frame of input to the video diffusion model are two reference views. Specifically, given sparse-view images $\mathcal{I} = \left\{ \boldsymbol{I}_{\text{ref}}^i \right\}_{i=1}^{K}$ as input, we aim to render consistent frames $f(\boldsymbol{I}_{\text{ref}}^{i-1}, \boldsymbol{I}_{\text{ref}}^i) = \{\boldsymbol{I}_{\text{ref}}^{i-1}, \boldsymbol{I}_2, ..., \boldsymbol{I}_T, \boldsymbol{I}_{\text{ref}}^i\} \in \mathbb{R}^{(T+2) \times 3 \times H \times W}$ where $T$ is the number of generated novel frames. To unify the notation, we denote the embedding of image condition in the pretrained video diffusion model as $F_g = g(\boldsymbol{I}_{\text{ref}})$ and the embedding of 3D structure condition as $F_{\mathcal{F}} = \mathcal{F}(\mathcal{P})$. Subsequently, we inject the 3D condition into the video diffusion process by interacting with the U-Net intermediate feature $F_{\text{in}}$ through the cross-attention of spatial layers:

$$F_{\text{out}} = \text{Softmax}(\frac{QK_g^T}{\sqrt{d}})V_g + \lambda_{\mathcal{F}} \cdot \text{Softmax}(\frac{QK_{\mathcal{F}}^T}{\sqrt{d}})V_{\mathcal{F}}, \tag{4}$$

where $Q = F_{\text{in}}W_Q, K_g = F_g W_K, V_g = F_g W_V, K_{\mathcal{F}} = F_{\mathcal{F}} W_K', V_{\mathcal{F}} = F_{\mathcal{F}} W_V'$ are the query, key, and value of 2D and 3D embeddings respectively. $W_Q, W_K, W_K', W_V, W_V'$ are the projection matrices and $\lambda_{\mathcal{F}}$ denotes the coefficient that balances image-conditioned and 3D structure-conditioned features. Given the first and last two views condition $c_{\text{view}}$ from $F_g$ and 3D structure condition $c_{\text{struc}}$ from $F_{\mathcal{F}}$, we apply the classifier-free guidance (Ho & Salimans, 2022) strategy to incorporate the condition and our training objective is:

$$\mathcal{L}_{\text{diffusion}} = \mathbb{E}_{\boldsymbol{x} \sim p, \epsilon \sim \mathcal{N}(0,I), t} \left[ \|\epsilon - \epsilon_\theta \left( \boldsymbol{x}_t, t, c_{\text{view}}, c_{\text{struc}} \right)\|_2^2 \right], \tag{5}$$

where $\boldsymbol{x}_t$ is the noise latent from the ground-truth views of the training data.

### 4.4 Confidence-Aware 3DGS Optimization.

Built upon the well-designed 3D structure condition, our video diffusion model generates highly consistent video frames, which can be used to reconstruct the 3D scene. As conventional 3D reconstruction methods are originally designed to handle real-captured photographs with calibrated camera metrics, directly applying these approaches to the generated videos is not effective to recover the coherent scene due to the uncertainty of unconstrained images (Wang et al., 2024a; Fan et al., 2024). To alleviate the uncertainty issue, we adopt a confidence-aware 3DGS mechanism to reconstruct the intricate scene. Different from recent approaches (Martin-Brualla et al., 2021a;

Ren et al., 2024) which model the uncertainty in per-image, we instead focus on a global alignment among a series of frames. For the generated frames $\left\{ \boldsymbol{I}^i \right\}_{i=1}^{K'}$, we denote $\hat{C}_i$ and $C_i$ as the per-pixel color value for predicted and generated view $i$. Then, we model the pixel values as a Gaussian distribution in our 3DGS, where the mean and variance of $\boldsymbol{I}^i$ are $C_i$ and $\sigma_i$. The variance $\sigma_i$ measures the discrepancy between the predicted and generated images. The uncertainty metric $\sigma_i$ for each image is estimated by minimizing the following negative log-likelihood among all frames:

$$\mathcal{L}_{I_i} = -\log \left( \frac{1}{\sqrt{2\pi\sigma_i^2}} \exp \left( -\frac{\|\hat{C}_i - C_i'\|_2^2}{2\sigma^2} \right) \right). \qquad (6)$$

where $C_i' = \mathcal{A}(C_i, \{C_i\}_{i=1}^{K'} \setminus C_i)$ and $\mathcal{A}$ is a tailored global align function to establish connections between each frame and the other frames, enabling a more robust global uncertainty estimation. Specifically, the training objective of DUSt3R is to map image pairs to 3D space, while the confidence map $\mathcal{C}$ represents the model's confidence in the pixel matches of image pairs within the 3D scene. Through its training process, DUSt3R inherently assigns low confidence to mismatched regions in image pairs, achieving the goal of Eq. 6. The confidence maps $\{\mathcal{C}_i\}_{i=1}^{K'}$ for each generated frames $\left\{ \boldsymbol{I}^i \right\}_{i=1}^{K'}$ are equivalent to the uncertainty $\sigma_i$. Meanwhile, the pairwise matching between all frames accomplishes the global alignment operation $\mathcal{A}$. Moreover, we introduce the LPIPS (Zhang et al., 2018b) loss to remove the artifacts and further enhance the visual quality. Towards this end, we formulate the confidence-aware 3DGS loss between the Gaussian rendered image $\hat{\boldsymbol{I}}^i$ and generated frame $\boldsymbol{I}^i$ as:

$$\mathcal{L}_{\text{conf}} = \sum_{i=1}^{K'} \mathcal{C}_i \left( \lambda_{\text{rgb}} \mathcal{L}_1(\hat{\boldsymbol{I}}^i, \boldsymbol{I}^i) + \lambda_{\text{ssim}} \mathcal{L}_{\text{ssim}}(\hat{\boldsymbol{I}}^i, \boldsymbol{I}^i) + \lambda_{\text{lpips}} \mathcal{L}_{\text{lpips}}(\hat{\boldsymbol{I}}^i, \boldsymbol{I}^i) \right). \qquad (7)$$

where $\mathcal{L}_1$, $\mathcal{L}_{\text{ssim}}$, and $\mathcal{L}_{\text{lpips}}$ denote the $L_1$, SSIM, and LPIPS loss, respectively, with $\lambda_{\text{rgb}}$, $\lambda_{\text{ssim}}$, and $\lambda_{\text{lpips}}$ being their corresponding coefficient parameters. In comparison to the photometric loss (*e.g.*, $\mathcal{L}_1$ and $\mathcal{L}_{\text{ssim}}$), the LPIPS loss mainly focuses on the high-level semantic information.

## 5 EXPERIMENTS

In this section, we conduct extensive experiments to evaluate our sparse-view reconstruction framework ReconX. We first present the setup of the experiment (Sec 5.1). Then we report our qualitative and quantitative results compared to feed-forward based methods (Sec 5.2) and per-scene optimization-based methods (Sec 5.3) in various settings. Finally, we conduct ablation studies to further verify the efficacy of our framework design (Sec 5.4). Please refer to our supplementary materials for more comparisons and visualizations.

### 5.1 EXPERIMENT SETUP

**Implementation Details.** In our framework, we choose DUSt3R (Wang et al., 2024a) as our unconstrained stereo 3D reconstruction backbone and the I2V model DynamiCrafter (Xing et al., 2023) (@ $512 \times 512$ resolution) as the video diffusion backbone. We first finetune the image cross-attention layers with 2000 steps on the learning rate $1 \times 10^{-4}$ for warm-up. Then we incorporate the 3D structure condition $c_{\text{struc}}$ into the video diffusion model and further finetune the spatial layers with 30K steps on the learning rate of $1 \times 10^{-5}$. Our video diffusion was trained on 3D scene datasets by sampling 32 frames with dynamic FPS at the resolution of $512 \times 512$ in a batch. The AdamW (Loshchilov & Hutter, 2017) optimizer is employed for optimization. At the inference of our video diffusion, we adopt the DDIM sampler (Song et al., 2022) using multi-condition classifier free guidance (Ho & Salimans, 2022). Similar to Xing et al. (2023), we adopt tanh gating to learn $\lambda_{\mathcal{F}}$ adaptively. The training is conducted on 8 NVIDIA A800 (80G) GPUs in two days. In the 3DGS optimization stage, we choose the point maps of the first and end frames as the initial global point cloud and all 32 generated frames are used to reconstruct the scene. Our implementation follows the pipeline of the original 3DGS (Kerbl et al., 2023), but unlike this method, we omit the adaptive control process and attain high-quality renderings in just 1000 steps. The coefficients $\lambda_{\text{rgb}}$, $\lambda_{\text{ssim}}$, and $\lambda_{\text{lpips}}$ are set to 0.8, 0.2, and 0.5, respectively.

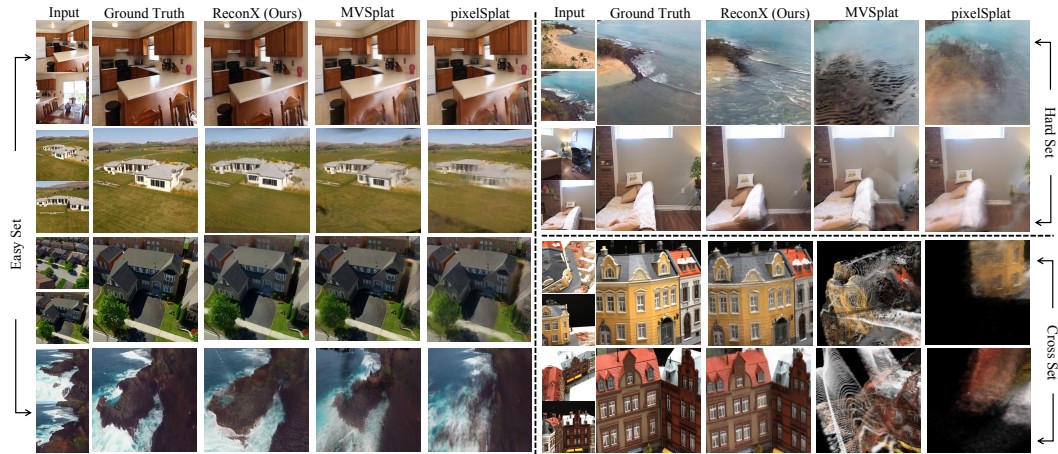

Figure 3: **Qualitative comparison with feed-forward based methods.** We provide the comparison of our ReconX with other baselines in Easy Set, Hard Set, and Cross Set. In comparison to these feed-forward based methods, ReconX achieves better visual quality and generalization.

**Datasets.** The video diffusion model of ReconX is trained on three datasets: RealEstate-10K (Zhou et al., 2018), ACID (Liu et al., 2021), and DL3DV-10K (Ling et al., 2024) based on the pretrained model. RealEstate-10K is a dataset downloaded from YouTube, which is split into 67,477 training scenes and 7,289 test scenes. The ACID dataset consists of natural landscape scenes, with 11,075 training scenes and 1,972 testing scenes. DL3DV-10K is a large-scale outdoor dataset containing 10,510 videos with consistent capture standards. For each scene video, we randomly sample 32 contiguous frames with random skips and serve the first and last frames as the input for our video diffusion model. To further validate our strong generalizability, we also directly evaluate our method on the DTU (Jensen et al., 2014), NeRF-LLFF (Mildenhall et al., 2019), and more challenging outdoor datasets Mip-NeRF 360 (Barron et al., 2022) and Tank-and-Temples dataset (Knapitsch et al., 2017).

**Baselines and Metrics.** To comprehensively demonstrate our strong capability in sparse-view reconstruction, we compare our ReconX with (a) feed-forward based methods trained from 3D scenes to learn 3D prior and (b) per-scene optimization based methods with specific priors (*e.g.,* , depth) for sparse-view reconstruction. Specifically, we compare with NeRF-based pixelNeRF (Yu et al., 2021) and MuRF (Xu et al., 2024); Light Field based GPNR (Suhail et al., 2022) and AttnRend (Du et al., 2023); and the recent state-of-the-art 3DGS-based pixelSplat (Charatan et al., 2024) and MVSplat (Chen et al., 2024a) in feed-forward based comparisons. On the other hand, we compare with SparseNeRF (Wang et al., 2023), original 3DGS (Kerbl et al., 2023), and DNGaussian (Li et al., 2024) for per-scene optimization comparisons. Furthermore, we qualitatively compare our method with more recent works CAT3D (Gao et al., 2024) and ReconFusion (Wu et al., 2024b) that incorporate generative power. For quantitative results, we report the standard metrics in NVS, including PSNR, SSIM (Wang et al., 2004), LPIPS (Zhang et al., 2018b).

## 5.2 COMPARISON WITH FEED-FORWARD BASED BASELINES

**Comparison for small angle variance in input views.** For fair comparison with baseline methods like MuNeRF (Xu et al., 2024), pixelSplat (Charatan et al., 2024), and MVSplat (Chen et al., 2024a), we first compare our reconX with baseline method from sparse views with small angle variance (see Easy Set from Table 1 and Figure 3). We observe that our ReconX surpasses all previous state-of-the-art models in terms of all metrics on visual quality and qualitative perception.

**Comparison for large angle variance in input views.** As MVSplat and pixelSplat are much better than previous baselines, we conduct thorough comparisons with them in more difficult settings. In more challenging settings (*i.e.,* given sparse views with large angle variance), our proposed ReconX demonstrate more significant improvement than baselines, especially in unseen and generalized viewpoints (see Hard Set from Table 2 and Figure 3). This clearly shows the effectiveness

| Easy Set | RealEstate10K | | | ACID | | |
|---|---|---|---|---|---|---|
| Method | PSNR ↑ | SSIM ↑ | LPIPS ↓ | PSNR ↑ | SSIM ↑ | LPIPS ↓ |
| pixelNeRF | 20.43 | 0.589 | 0.550 | 20.97 | 0.547 | 0.533 |
| GPNR | 24.11 | 0.793 | 0.255 | 25.28 | 0.764 | 0.332 |
| AttnRend | 24.78 | 0.820 | 0.213 | 26.88 | 0.799 | 0.218 |
| MuRF | 26.10 | 0.858 | 0.143 | 28.09 | 0.841 | 0.155 |
| pixelSplat | 25.89 | 0.858 | 0.142 | 28.14 | 0.839 | 0.150 |
| MVSplat | 26.39 | 0.839 | 0.128 | 28.25 | 0.843 | 0.144 |
| **ReconX** | **28.31** | **0.912** | **0.088** | **28.84** | **0.891** | **0.101** |

Table 1: **Quantitative comparisons with feed-forward based methods** for small angle variance (Easy Set) in input views. For each scene, the model takes two views as input and renders three novel views for evaluation.

| Hard Set | ACID | | | RealEstate10K | | |
|---|---|---|---|---|---|---|
| Method | PSNR ↑ | SSIM ↑ | LPIPS ↓ | PSNR ↑ | SSIM ↑ | LPIPS ↓ |
| pixelSplat | 16.83 | 0.476 | 0.494 | 19.62 | 0.730 | 0.270 |
| MVSplat | 16.49 | 0.466 | 0.486 | 19.97 | 0.732 | 0.245 |
| **ReconX** | **24.53** | **0.847** | **0.083** | **23.70** | **0.867** | **0.143** |
| **Cross Set** | LLFF | | | DTU | | |
| pixelSplat | 16.83 | 0.476 | 0.494 | 19.62 | 0.730 | 0.270 |
| MVSplat | 16.49 | 0.466 | 0.486 | 19.97 | 0.732 | 0.245 |
| **ReconX** | **24.53** | **0.847** | **0.083** | **23.70** | **0.867** | **0.143** |

Table 2: **Quantitative comparison with feed-forward based methods** for large angle variance (Hard Set) in input views and cross-dataset (Cross Set) comparisons to evaluate generalization ability.

| Method | 2-view | | | 3-view | | | 6-view | | | 9-view | | |
|---|---|---|---|---|---|---|---|---|---|---|---|---|
| | PSNR↑ | SSIM↑ | LPIPS↓ | PSNR↑ | SSIM↑ | LPIPS↓ | PSNR↑ | SSIM↑ | LPIPS↓ | PSNR↑ | SSIM↑ | LPIPS↓ |
| **Mip-NeRF 360** | | | | | | | | | | | | |
| 3DGS | 10.36 | 0.108 | 0.776 | 10.86 | 0.126 | 0.695 | 12.48 | 0.180 | 0.654 | 13.10 | 0.191 | 0.622 |
| SparseNeRF | 11.47 | 0.190 | 0.716 | 11.67 | 0.197 | 0.718 | 14.79 | 0.150 | 0.662 | 14.90 | 0.156 | 0.656 |
| DNGaussian | 10.81 | 0.133 | 0.727 | 11.13 | 0.153 | 0.711 | 12.20 | 0.218 | 0.688 | 13.01 | 0.246 | 0.678 |
| **ReconX (Ours)** | **13.37** | **0.283** | **0.550** | **16.66** | **0.408** | **0.427** | **18.72** | **0.451** | **0.390** | **18.17** | **0.446** | **0.382** |
| **Tank and Temples** | | | | | | | | | | | | |
| 3DGS | 9.57 | 0.108 | 0.779 | 10.15 | 0.118 | 0.763 | 11.48 | 0.204 | 0.685 | 12.50 | 0.202 | 0.669 |
| SparseNeRF | 9.23 | 0.191 | 0.632 | 9.55 | 0.216 | 0.633 | 12.24 | 0.274 | 0.615 | 12.74 | 0.294 | 0.608 |
| DNGaussian | 10.23 | 0.156 | 0.643 | 11.25 | 0.204 | 0.584 | 12.92 | 0.231 | 0.535 | 13.01 | 0.256 | 0.520 |
| **ReconX (Ours)** | **14.28** | **0.394** | **0.564** | **15.38** | **0.437** | **0.483** | **16.27** | **0.497** | **0.420** | **18.38** | **0.556** | **0.355** |
| **DL3DV** | | | | | | | | | | | | |
| 3DGS | 9.46 | 0.125 | 0.732 | 10.97 | 0.248 | 0.567 | 13.34 | 0.332 | 0.498 | 14.99 | 0.403 | 0.446 |
| SparseNeRF | 9.14 | 0.137 | 0.793 | 10.89 | 0.214 | 0.593 | 12.15 | 0.234 | 0.577 | 12.89 | 0.242 | 0.576 |
| DNGaussian | 10.10 | 0.149 | 0.523 | 11.10 | 0.274 | 0.577 | 12.65 | 0.330 | 0.548 | 13.46 | 0.367 | 0.541 |
| ReconX (Ours) | **13.60** | **0.307** | **0.554** | **14.97** | **0.419** | **0.444** | **17.45** | **0.476** | **0.426** | **18.59** | **0.584** | **0.386** |

Table 3: **Quantitative comparisons with per-scene optimization based methods** on MipNeRF 360 and Tank and Temples, and DL3DV. We evaluate the reconstruction performance with different input views for each scene.

of ReconX in creating more consistent observations from video diffusion to mitigate the inherent ill-posed sparse-view reconstruction problem.

**Cross-dataset generalization.** Unleashing the strong generative power of the video diffusion model through 3D structure condition, our ReconX is inherently superior in generalizing to out-of-distribution novel scenes. To demonstrate the strong generalizability of ReconX, we conduct two cross-dataset evaluations. For a fair comparison, we train the models solely on the RealEstate10K and directly test them on two popular NVS datasets (*i.e.,* NeRF-LLFF (Mildenhall et al., 2019) and DTU (Jensen et al., 2014)). As shown in Cross Set from Table 2 and Figure 3, the competitive baseline methods MVSplat (Chen et al., 2024a) and pixelSplat (Charatan et al., 2024) fail to render such OOD datasets which contain different camera distributions and image appearance, leading to dramatic performance degradation. In contrast, our ReconX shows impressive generalizability and the gain is larger when the domain gap from training and test data becomes larger.

**Assessing more-view quality.** ReconX is agnostic to the number of input views. Specifically, given $N$ views as input, we sample a plausible camera trajectory to render image pairs using our video diffusion models and finally optimize the 3D scene from all generated frames. For a fair comparison with Chen et al. (2024a), we verify this by testing on DTU with three context views. Our results are PSNR: 22.83, SSIM: 0.512, LPIPS: 0.317, MVSplat's are PSNR: 14.30, SSIM: 0.508, LPIPS: 0.371, and pixelSplat's are PSNR: 12.52, SSIM: 0.367, LPIPS: 0.585. Compared to the two-view results (Table 2), our ReconX and MVSplat both achieve better performance given more input views while we are much better than MVSplat. However, pixelSplat performs worse when using more views also shown in Chen et al. (2024a).

### 5.3 COMPARISON WITH PER-SCENE OPTIMIZATION BASED BASELINES

To verify the capability of ReconX in sparse-view reconstruction in more challenging outdoor settings, we compare with per-scene optimization-based methods in different input views (*i.e.,* , 2, 3,

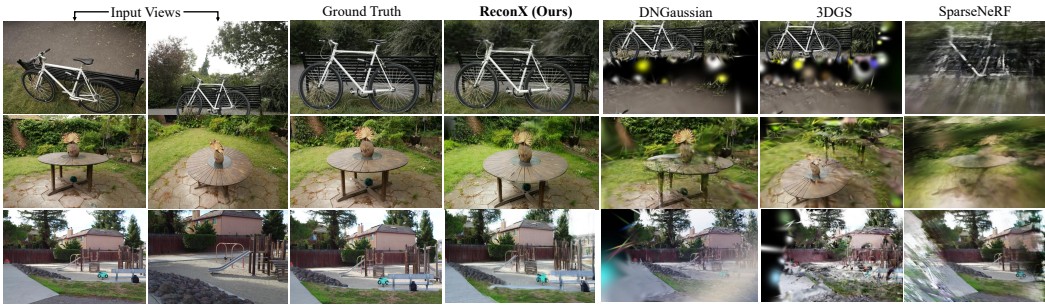

Figure 4: **Qualitative comparison with per-scene optimization based methods** on Mip-Nerf 360 and Tank and Temples. With two sparse views as input, our ReconX achieves much better reconstruction quality compared with baselines.

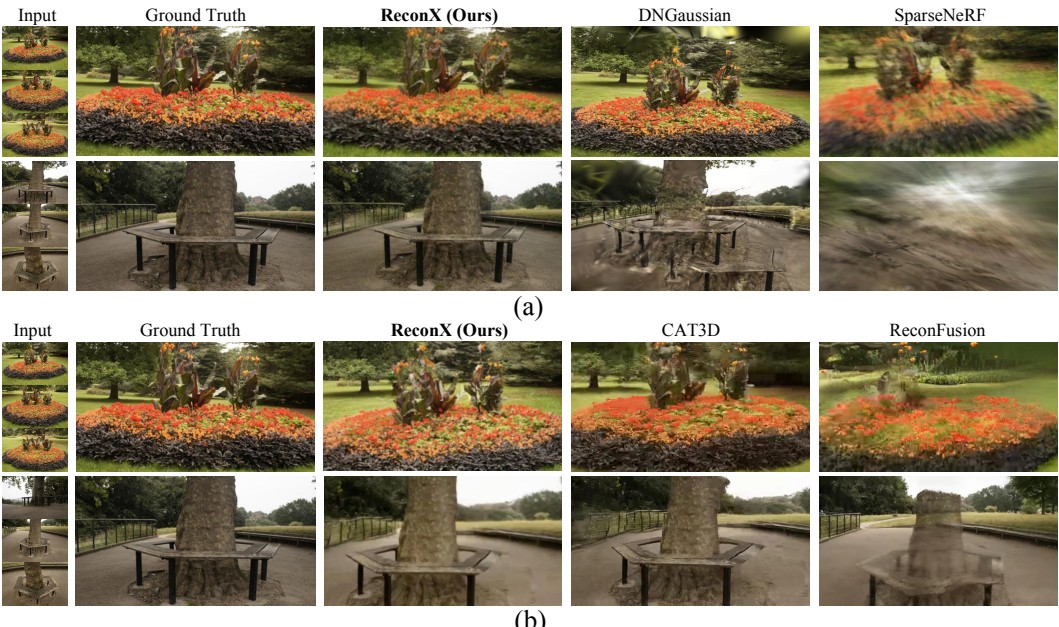

Figure 5: **More qualitative comparisons** with three sparse views as input. (a) comparison with DNGaussian and SparseNeRF with random selected input views. (b) comparison with CAT3D and ReconFusion on input views selected with heuristic loss (Wu et al., 2024b).

6, and 9 views) in Table 3 and more visual comparisons in Figure 4. We observe that our method outperforms all the other per-scene optimization baselines in PSNR, SSIM, and LPIPS scores. As shown in Figure 4, we find that the baselines produce extremely blurry results in only two view settings with noisy camera estimations. In contrast, by unleashing the generative power of the video diffusion model, our ReconX can create more observations from only two sparse views and ensures high-quality novel view rendering, avoiding local minima issues. To further demonstrate our superiority, we compare in 3-view setting even with recent works CAT3D (Gao et al., 2024) and ReconFusion (Wu et al., 2024b) that incorporate generative prior to mitigate ill-posed sparse view reconstruction in Table 5 and Figure 5. Since the codes for CAT3D and ReconFusion are not available, we downloaded the results directly from the project pages using three input views as provided in their papers. The results show that ReconX can produce higher-frequency details in novel views.

## 5.4 Ablation Study and Analysis

We carry out ablation studies on RealEstate10K to analyze the design of our ReconX framework in Table 4 and Figure 6. A naive combination of pretrained video diffusion model and Gaussian

| Video diffusion | 3D structure condition | DUSt3R init | confidence-aware opt. | LPIPS loss | PSNR↑ | SSIM↑ | LPIPS↓ |
|---|---|---|---|---|---|---|---|
| - | - | ✓ | - | - | 17.34 | 0.527 | 0.259 |
| ✓ | - | ✓ | - | - | 19.70 | 0.789 | 0.229 |
| ✓ | - | ✓ | ✓ | ✓ | 25.13 | 0.901 | 0.131 |
| ✓ | ✓ | - | ✓ | ✓ | 27.11 | 0.908 | 0.113 |
| ✓ | ✓ | ✓ | - | ✓ | 27.83 | 0.897 | 0.097 |
| ✓ | ✓ | ✓ | ✓ | - | 27.47 | 0.906 | 0.111 |
| ✓ | ✓ | ✓ | ✓ | ✓ | **28.31** | **0.912** | **0.088** |

Table 4: **Quantitative results of ablation study**. We report the quantitative metrics in ablations of our framework in real-world data (Zhou et al., 2018).

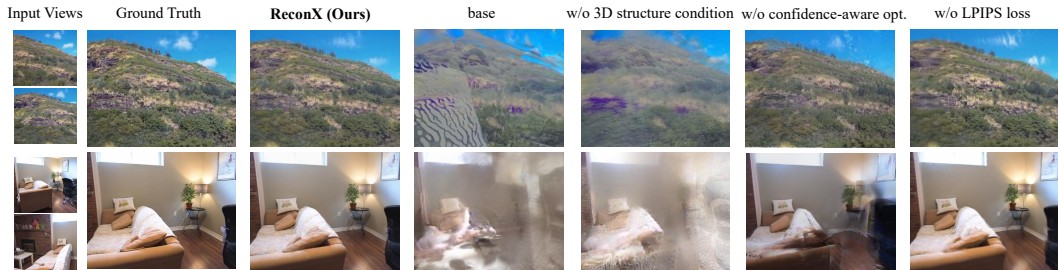

Figure 6: **Visualization results of ablation study.** We ablate the design choices of 3D structure guidance, confidence-aware optimization, and the LPIPS loss.

Splatting is regarded as the "base". Specifically, we ablate on the following aspects of our method: 3D structure condition, DUSt3R initialization, confidence-aware optimization, and LPIPS loss. The results indicate that the omission of any of these elements leads to a degradation in terms of quality and consistency. Notably, the basic combination of original video diffusion model and 3DGS leads to significant distortion of the scene. The absence of 3D structure condition causes inconsistent generated frames especially in distant input views, resulting in blur and artifact issues. The lack of confidence-aware optimization leads to suboptimal results in some local detail areas. Adding LPIPS loss in confidence-aware 3DGS optimization would provide clearer rendering views. Moreover, we ablate the impact of DUSt3R and video diffusion priors in Figure 7. Although the point cloud may not include enough high-quality information, such coarse 3D structure is sufficient to guide the video diffusion in our ReconX to fill in the distortions, occlusions or missing regions. This demonstrates that our ReconX has learned a comprehensive understanding of the 3D scene and can generate high-quality novel views from imperfect conditional information and exhibit robustness to the point cloud conditions. This illustrates the effectiveness of our overall framework (Figure 2), which drives generalizable and high-fidelity 3D reconstruction given only sparse views as input.

# 6 CONCLUSION

In this paper, we introduce ReconX, a novel sparse-view 3D reconstruction framework that reformulates the inherently ambiguous reconstruction problem as a generation problem. The key to our success is that we unleash the strong prior of video diffusion models to create more plausible observations frames for sparse-view reconstruction. Grounded by the empirical study and theoretical analysis, we propose to incorporate 3D structure guidance into the video diffusion process for better 3D consistent video frames generation. What's more, we propose a 3D confidence-aware scheme to optimize the final 3DGS from generated frames, which effectively addresses the uncertainty issue. Extensive experiments demonstrate the superiority of our ReconX over the latest state-of-the-art methods in terms of high quality and strong generalizability in unseen data.

**Limitations and Future Work.** Although ReconX achieves remarkable reconstruction results in novel viewpoints, the quality still seems to be limited by the backbone as we choose the U-Net based diffusion model DynamiCrafter (Xing et al., 2023). We expect that this issue can be solved with open-sourced larger video diffusion models (*e.g.,* DiT-based framework). In the future, it is interesting to integrate 3DGS optimization directly with the video generation model, enabling more efficient end-to-end 3D scene reconstruction. We are also interested in exploring consistent 4D scene reconstruction. We believe that ReconX provides a promising research direction to craft intricate 3D worlds from video diffusion models and hope it will inspire more works in the future.

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

## A  PRELIMINARIES

**Video Diffusion Models.** Diffusion models (Ho et al., 2020; Song et al., 2020) have emerged as the cutting-edge paradigm to generate high-quality videos. These models learn the underlying data distribution by adding and removing noise on the clean data. The forward process aims to transform a clean data sample $\boldsymbol{x}_0 \sim p(\boldsymbol{x})$ to a pure Gaussian noise $\boldsymbol{x}_T \sim \mathcal{N}(0, I)$, following the process:

$$\boldsymbol{x}_t = \sqrt{\bar{\alpha}_t}\boldsymbol{x}_0 + \sqrt{1 - \bar{\alpha}_t}\epsilon, \quad \epsilon \sim \mathcal{N}(\boldsymbol{0}, \boldsymbol{1}), \tag{8}$$

where $\boldsymbol{x}_t$ and $\bar{\alpha}_t$ denotes the noisy data and noise strength at the timestep $t$. The denoising neural network $\epsilon_\theta$ is trained to predict the noises added in the forward process, which is achieved by the MSE loss:

$$\mathcal{L} = \mathbb{E}_{\boldsymbol{x} \sim p, \epsilon \sim \mathcal{N}(0, I), c, t}\left[\|\epsilon - \epsilon_\theta\left(\boldsymbol{x}_t, t, c\right)\|_2^2\right], \tag{9}$$

where $c$ represents the embeddings of conditions like text or image prompt. For the video diffusion models, Latent Diffusion Models (LDMs) (Rombach et al., 2022), which compress images into the latent space, are commonly employed to mitigate the computation complexity while maintaining competitive performance.

**3D Gaussian Splatting.** 3DGS (Kerbl et al., 2023) represents a scene explicitly by utilizing a set of 3D Gaussian spheres, achieving a fast and high-quality rendering. A 3D Gaussian is modeled by a position vector $\boldsymbol{\mu} \in \mathbb{R}^3$, a covariance matrix $\boldsymbol{\Sigma} \in \mathbb{R}^{3 \times 3}$, an opacity $\alpha \in \mathbb{R}$, and spherical harmonics (SH) coefficient $\boldsymbol{c} \in \mathbb{R}^k$ (Ramamoorthi & Hanrahan, 2001). Moreover, the Gaussian distribution is formulated as the following:

$$G(x) = e^{-\frac{1}{2}(x-\boldsymbol{\mu})^T \boldsymbol{\Sigma}^{-1}(x-\boldsymbol{\mu})}, \tag{10}$$

where $\boldsymbol{\Sigma} = \boldsymbol{R}\boldsymbol{S}\boldsymbol{S}^T\boldsymbol{R}^T$, $\boldsymbol{S}$ denotes the scaling matrix and $\boldsymbol{R}$ is the rotation matrix.

In the rendering stage, the 3D Gaussian spheres are transformed into 2D camera planes through rasterization (Zwicker et al., 2001). Specifically, given the perspective transformation matrix $\boldsymbol{W}$ and Jacobin of the projection matrix $\boldsymbol{J}$, the 2D covariance matrix in the camera space is computed as

$$\boldsymbol{\Sigma}^{'} = \boldsymbol{J}\boldsymbol{W}\boldsymbol{\Sigma}\boldsymbol{W}^T\boldsymbol{J}^T. \tag{11}$$

For every pixel, the Gaussians are traversed in depth order from the image plane, and their view-dependent colors $c_i$ are combined through alpha compositing, leading to the pixel color $C$:

$$C = \sum_{i \in N} c_i \alpha_i \prod_{j=1}^{i-1} (1 - \alpha_i). \tag{12}$$

**End-to-end Dense Unconstrained Stereo.** DUSt3R (Wang et al., 2024a) is a new model to predict a dense and accurate 3D scene representation solely from image pairs without any prior information about the scene. Given two unposed images $\{\boldsymbol{I}_1, \boldsymbol{I}_2\}$, this end-to-end model is trained to estimate the point maps $\{P_{1,1}, P_{2,1}\}$ and confidence maps $\{\mathcal{C}_{1,1}, \mathcal{C}_{2,1}\}$, which can be utilized to recover the camera parameters and dense point cloud. The training procedure for view $v \in \{1, 2\}$ is formulated as a regression loss:

$$\mathcal{L} = \left\| \frac{1}{z_i} \cdot P_{v,1} - \frac{1}{\hat{z}_i} \cdot \hat{P}_{v,1} \right\|, \tag{13}$$

where $P$ and $\hat{P}$ denote the ground-truth and prediction point maps, respectively. The scaling factors $z_i = \text{norm}(P_{1,1}, P_{2,1})$ and $\hat{z}_i = \text{norm}(\hat{P}_{1,1}, \hat{P}_{2,1})$ are adopted to normalize the point maps, which merely indicate the mean distance $D$ of all valid points from the origin:

$$\text{norm}\left(P_{1,1}, P_{2,1}\right) = \frac{1}{|D_1| + |D_2|} \sum_{v \in \{1,2\}} \sum_{i \in D_v} \left\|P_v^i\right\|. \tag{14}$$

## B  MORE IMPLEMENTATION DETAILS

**Implementation of PosEmb.** The PosEmb implemented in our paper is a column-wise positional embedding function: $\mathbb{R}^3 \to \mathbb{R}^C$, where $C$ is the dimension of embedding. More specifically, the

PosEmb function is implemented as follows: (1) Fixed Sinusoidal Basis: The basis $\mathbf{e}$ is a 3D sinusoidal encoding: $\mathbf{e} = [\sin(2^0 \pi p), \sin(2^1 \pi p), \dots]$, where $p \in \mathbb{R}^3$ is the position. (2) Embedding Calculation: The input $\mathbf{x}$ is projected onto $\mathbf{e}$ and its sine and cosine are concatenated: $\mathbf{embeddings} = \text{concat}(\sin(\mathbf{proj}), \cos(\mathbf{proj}))$. (3) Learnable Transformation: The positional encoding is passed through an MLP along with the input $\mathbf{x}$: $\mathbf{y} = \text{MLP}(\text{concat}(\mathbf{embeddings}, \mathbf{x}))$. In short, PosEmb combines a fixed sinusoidal encoding with a learnable MLP transformation.

**More details of transformer-based encoder.** For the transformer-based encoder, we encode the DUSt3R point cloud data to a fixed-length sparse representation of the point cloud. Specifically, we first employ a subsampling based on farthest point sampling (FPS) to reduce the point cloud to a smaller set of key points while retaining its overall structural characteristics. Then, we apply cross-attention between the embeddings of the original point cloud and downsampled point cloud. This mechanism can be interpreted as a form of partial self attention, where the downsampled points act as query anchors that aggregate information from the original point cloud. The encoder is not initialized from any pretrained models. Instead, it is trained jointly with the video diffusion model in an end-to-end manner. This design choice ensures that the encoder is specifically adapted to the characteristics of DUSt3R point clouds in our experiment datasets.

**Camera alignment.** We consider that the camera pose from DUSt3R is not aligned with the COLMAP cameras. Since we adopt the point cloud and camera poses from DUSt3R in our experiment, it is necessary for us to unify the training and testing images into the same DUSt3R coordinate system. Specifically, we process the training and testing images together through DUSt3R to obtain the corresponding camera poses, and utilize only the point maps from the training set as the initial point cloud for optimizing 3DGS.

**Test view selection.** In comparison with feed-forward based methods, we follow MVSplat (Chen et al., 2024a) and pixelSplat (Charatan et al., 2024) to choose test views in Easy Set. For Hard Set, we choose the frame intervals much larger (i.e., $> 200$ frames) than Easy Set. For Tank-and-Templates and DL3DV datasets, we select the training views evenly from all the frames and use every 8th of the remaining frames for evaluation. For nine scenes in Mip-NeRF 360 dataset, we manually choose a training 9-view split of views that are uniformly distributed around the hemisphere and pointed toward the central object of interest. Then we further choose the 6- and 3-view splits to be subsets of the 9-view split.

## C  THEORETICAL PROOF

**Proposition 1.** *Let $\boldsymbol{\theta}^*, \psi^* = g^*$ be the optimal solution of the solely image-based conditional diffusion scheme and $\tilde{\boldsymbol{\theta}}^*, \tilde{\psi}^* = \{g^*, \mathcal{F}^*\}$ be the optimal solution of diffusion scheme with native 3D prior. Suppose the divergence $\mathcal{D}$ is convex and the embedding function space $\Psi$ includes all measurable functions, we have $\mathcal{D}(q(\boldsymbol{x}) \| p_{\tilde{\boldsymbol{\theta}}^*, \tilde{\psi}^*}(\boldsymbol{x})) < \mathcal{D}(q(\boldsymbol{x}) \| p_{\boldsymbol{\theta}^*, \psi^*}(\boldsymbol{x})).$*

*Proof.* According to the convexity of $\mathcal{D}$ and Jensen's inequality $\mathcal{D}(\mathbb{E}[X]) \leq \mathbb{E}[\mathcal{D}(X)]$, where $X$ is a random variable, we have:

$$
\begin{aligned}
\mathcal{D}\Big(q(\boldsymbol{x}) \| p_{\tilde{\boldsymbol{\theta}}^*, \tilde{\psi}^*}(\boldsymbol{x})\Big) &= \mathcal{D}\Big(\mathbb{E}_{q(s)} q(\boldsymbol{x}|s) \| \mathbb{E}_{q(s)} p_{\tilde{\boldsymbol{\theta}}^*, \tilde{\psi}^*}(\boldsymbol{x}|s)\Big) \\
&\leq \mathbb{E}_{q(s)} \mathcal{D}\Big(q(\boldsymbol{x}|s) \| p_{\tilde{\boldsymbol{\theta}}^*, \tilde{\psi}^*}(\boldsymbol{x}|s)\Big) \\
&= \mathbb{E}_{q(s)} \mathcal{D}\Big(q(\boldsymbol{x}|s) \| p_{\tilde{\boldsymbol{\theta}}^*, g^*, \mathcal{F}^*}(\boldsymbol{x}|s)\Big),
\end{aligned}
\tag{15}
$$

where we incorporate an intermediate variable $s$, which represents a specific scene. $q(\boldsymbol{x}|s)$ indicates the conditional distribution of rendering data $\boldsymbol{x}$ given the specific scene $s$. According to the definition of $\tilde{\boldsymbol{\theta}}^*, g^*, \mathcal{F}^*$, we have:

$$
\begin{aligned}
\mathbb{E}_{q(s)} \mathcal{D}\Big(q(\boldsymbol{x}|s) \| p_{\tilde{\boldsymbol{\theta}}^*, g^*, \mathcal{F}^*}(\boldsymbol{x}|s)\Big) &= \min_{\boldsymbol{\theta}, g, \mathcal{F}} \mathbb{E}_{q(s)} \mathcal{D}\left(q(\boldsymbol{x}|s) \| p_{\boldsymbol{\theta}, g, \mathcal{F}}(\boldsymbol{x}|s)\right) \\
&= \min_{\boldsymbol{\theta}} \mathbb{E}_{q(s)} \min_{g(s), \mathcal{F}(s)} \mathcal{D}\left(q(\boldsymbol{x}|s) \| p_{\boldsymbol{\theta}, g(s), \mathcal{F}(s)}(\boldsymbol{x})\right) \\
&= \min_{\boldsymbol{\theta}} \mathbb{E}_{q(s)} \min_{g, E} \mathcal{D}\left(q(\boldsymbol{x}|s) \| p_{\boldsymbol{\theta}, g, E}(\boldsymbol{x})\right),
\end{aligned}
\tag{16}
$$

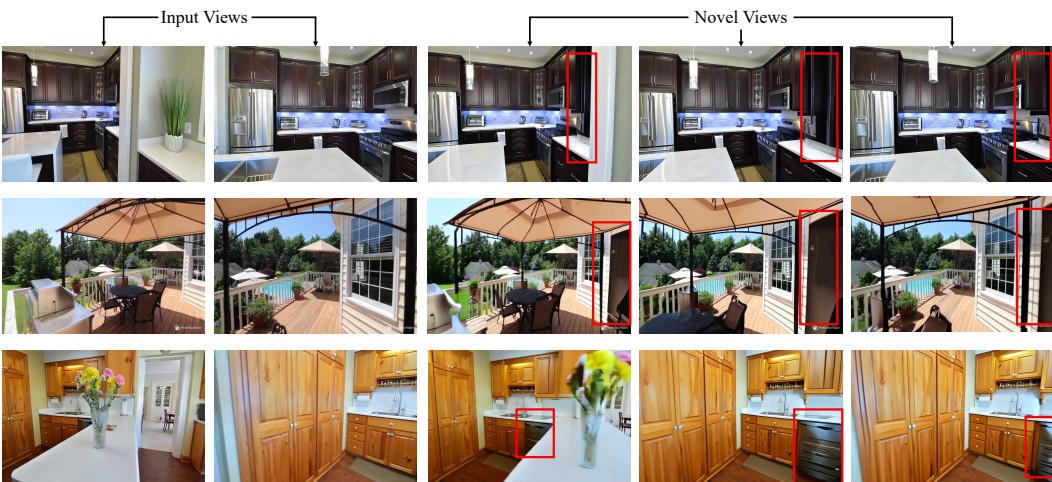

Figure 7: **Visualization results on the impact of video diffusion.** We ablate the impact of video diffusion in improving the reconstruction result of DUSt3R.

Figure 8: **Evaluation of extrapolation ability of ReconX.** We highlight the extrapolated regions in the red boxes in the novel rendered views.

where $E$ is the general 3D encoder in 3D structure conditional scheme while it is a redundant embedding in solely image-based conditional scheme, *i.e.,* $\psi = \{g, E(\varnothing)\}$. Combining Equation 15 and 16, we have:

$$
\begin{aligned}
\mathcal{D}\left(q(\boldsymbol{x})\|p_{\tilde{\boldsymbol{\theta}}^*,\tilde{\psi}^*}(\boldsymbol{x})\right) &\le \min_{\boldsymbol{\theta}} \mathbb{E}_{q(s)} \min_{g,E} \mathcal{D}\left(q(\boldsymbol{x}|s)\|p_{\boldsymbol{\theta},g,E}(\boldsymbol{x})\right) \\
&< \min_{\boldsymbol{\theta},g,E} \mathcal{D}\left(q(\boldsymbol{x})\|p_{\boldsymbol{\theta},g,E}(\boldsymbol{x})\right) = \min_{\boldsymbol{\theta},g,E(\varnothing)} \mathcal{D}\left(q(\boldsymbol{x})\|p_{\boldsymbol{\theta},g,E(\varnothing)}(\boldsymbol{x})\right) \quad (17) \\
&= \min_{\boldsymbol{\theta},\psi} \mathcal{D}\left(q(\boldsymbol{x})\|p_{\boldsymbol{\theta},\psi}(\boldsymbol{x})\right) = \mathcal{D}\left(q(\boldsymbol{x})\|p_{\boldsymbol{\theta}^*,\psi^*}(\boldsymbol{x})\right).
\end{aligned}
$$

The second inequality holds because given general real-world scene $s$ in any parameter $\boldsymbol{\theta} \in \Theta$, approximating $q(\boldsymbol{x}|s)$ is simpler than $q(\boldsymbol{x})$ by only tuning the encoder $E$ of $p_{\boldsymbol{\theta},g,E}$[1], *i.e.,* $\min_E \mathcal{D}\left(q(\boldsymbol{x}|s)\|p_{\boldsymbol{\theta},g,E}(\boldsymbol{x})\right) < \min_E \mathcal{D}\left(q(\boldsymbol{x})\|p_{\boldsymbol{\theta},g,E}(\boldsymbol{x})\right)$ holds almost everywhere (a.e.), representing $\mathcal{P}_{q(s)}\left\{\min_E \mathcal{D}\left(q(\boldsymbol{x}\mid s)\|p_{\boldsymbol{\theta},g,E}(\boldsymbol{x})\right) < \min_E \mathcal{D}\left(q(\boldsymbol{x})\|p_{\boldsymbol{\theta},g,E}(\boldsymbol{x})\right)\right\} = 1$ .

Consequently, the proof of Proposition 1 has been done.

## D   MORE RESULTS AND ANALYSIS

**Evaluation of our extrapolation ability.** As we use a pair of input views in our method, it is worthy to note that if the angular difference between the two views is too large, it is hard to ensure that the entire interpolated region falls within the visible perspective of the input views, which requires the extrapolation ability. We have evaluated it in our generalizable experiments with DTU dataset. For instance, in the case of DTU in Figure 3, we cannot see the roof area from the input views, while

---

[1]A simple verifiable case is to optimize the parameters of 3DGS by only 2D images (solely image-based conditional learning) or using a SFM initialization from collected images (native 3D conditional learning) before optimization. The latter provides a more constrained and optimal solution space.

| Method | 3-view | | | 6-view | | | 9-view | | |
|--------|--------|------|------|--------|------|------|--------|------|------|
| | PSNR↑ | SSIM↑ | LPIPS↓ | PSNR↑ | SSIM↑ | LPIPS↓ | PSNR↑ | SSIM↑ | LPIPS↓ |
| Zip-NeRF | 12.77 | 0.271 | 0.705 | 13.61 | 0.284 | 0.663 | 14.30 | 0.312 | 0.633 |
| ZeroNVS | 14.44 | 0.316 | 0.680 | 15.51 | 0.337 | 0.663 | 15.99 | 0.350 | 0.655 |
| ReconFusion | 15.50 | 0.358 | 0.585 | 16.93 | 0.401 | 0.544 | 18.19 | 0.432 | 0.511 |
| CAT3D | 16.62 | 0.377 | 0.515 | 17.72 | 0.425 | 0.482 | 18.67 | 0.460 | 0.460 |
| **ReconX (Ours)** | **17.16** | **0.435** | **0.407** | **19.20** | **0.473** | **0.378** | **20.13** | **0.482** | **0.356** |

Table 5: **Quantitative comparisons with more per-scene optimization based methods** on MipN-eRF 360. We evaluate the reconstruction performance with different input views for each scene.

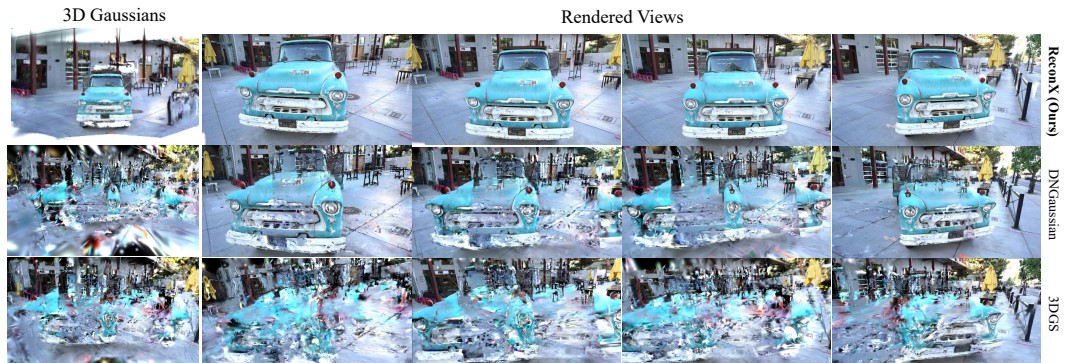

Figure 9: Rendering comparison with Gaussian-based methods frame by frame.

our ReconX is able to extrapolate and generate the red and yellow roof with 3D structure-guided generative prior. To further demonstrate the extrapolation capability of our method, we conduct a specific experiment in Figure 8. This experiment selects two views with large angular spans and highlights the extrapolated regions in the red boxes in the novel-rendered views. This emphasizes our model's generative power to extrapolate unseen regions and extend beyond the visible input views.

**More visual results in outdoor scenes.** Regarding the DL3DV dataset, we trained our model on this to demonstrate its performance on outdoor scenes. Due to the limitations of feed-forward methods on this dataset, we did not present quantitative results in the main paper, as these methods fail on it. However, to highlight our model's strengths in outdoor environments, we have included visual results in the supplementary video and have added comparisons with per-scene optimization methods in Table 3. We have also provided more visual results on DL3DV in Figure 12. We also compare our ReconX in 3D Gaussians with frame-by-frame results in Figure 9.

**More quantitative comparisons.** As the data is open-sourced in ReconFusion (Wu et al., 2024b) we conduct an additional quantitative experiment in comparison with ZipNeRF (Barron et al., 2023), ZeroNVS (Sargent et al., 2023), CAT3D (Gao et al., 2024), and ReconFusion (Wu et al., 2024b). It is worth noting that the data split used in CAT3D (Gao et al., 2024) follows a heuristic loss (Gao et al., 2024) to encourage reasonable camera spacing and coverage of the central object. We observe that our ReconX is better than all baselines in Table 5.

**More extrapolation ability discussion.** In the main paper, we focus on unleashing the video diffusion model to generate 3D consistent views through two-view interpolation. Specifically, we have conducted experiments to verify the capability of ReconX to recover 3D scenes from large angle variance in input views (see Figure 3 and Figure 8), showing its ability to extrapolate the occlusions and correct the inaccurate geometry details given the coarse 3D structure. As the position of our conditional images in ReconX is inherently flexible, allowing us to unleash more extrapolation capability by adjusting the placement of the conditional images.

To further investigate the generative capabilities of our framework and demonstrate its extrapolation potential, we conduct experiments by conditioning on the first and an intermediate frame of the target video with a new tuning version of the video diffusion model in ReconX by only moving

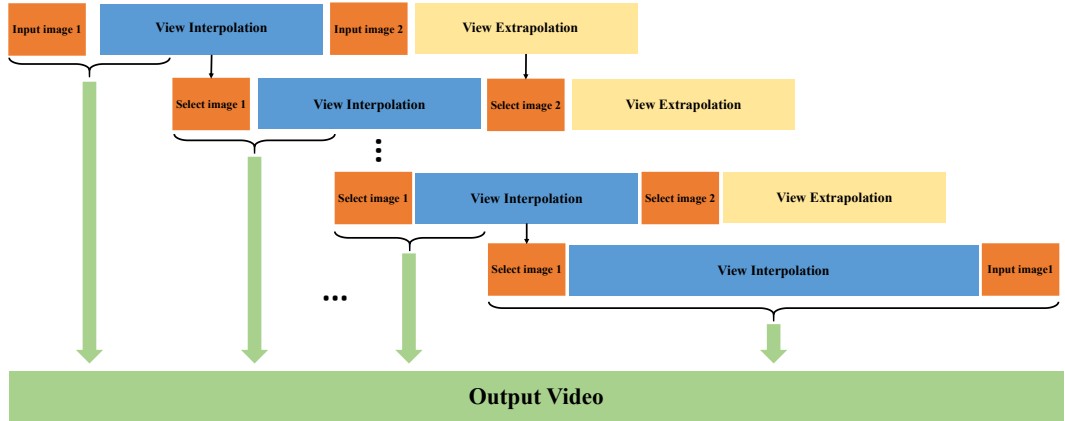

Figure 10: The incremental strategy to generate full 360-degree scenes using only two initial images.

the last frame to the intermediate position. In this setup, frames between the first and intermediate images correspond to view interpolation, while frames beyond the intermediate image correspond to extrapolation.

- View interpolation can not only synthesize visible areas between the input images but also generate previously unseen regions caused by occlusions.
- View extrapolation continues along the camera's motion trajectory, generating entirely new content not present in the input images, such as unseen objects and expanded scene regions.

Such extrapolation ability allows us to even recover a 360-degree scene from only two sparse views. Specifically, we adopt an incremental generation approach shown in Figure 10. Given two initial input images (*i.e.,* input image 1 and input image 2 in Figure 10), we first generate a video sequence divided into four segments: input image 1, view interpolation, input image 2, and view extrapolation. From this generated frame sequence, we select two images—one from the interpolation part and another one from the extrapolation part. These two images function as a sliding window, and repeat the generation process, progressively advancing with each iteration. This approach allows our framework to autoregressively generate a much longer 360-degree panoramic sequence while maintaining a limited-length video frame window. In the final iteration, we select one image from the video generated in the previous step (*i.e.,* select image 1 in Figure 10) and pair it with the original first input image (*i.e.,* input image 1 in Figure 10) as the input pair. This ensures a seamless connection back to the starting view, completing a full 360-degree scene reconstruction shown in Figure 11. This incremental approach demonstrates the strong generative extrapolation potential of our method.

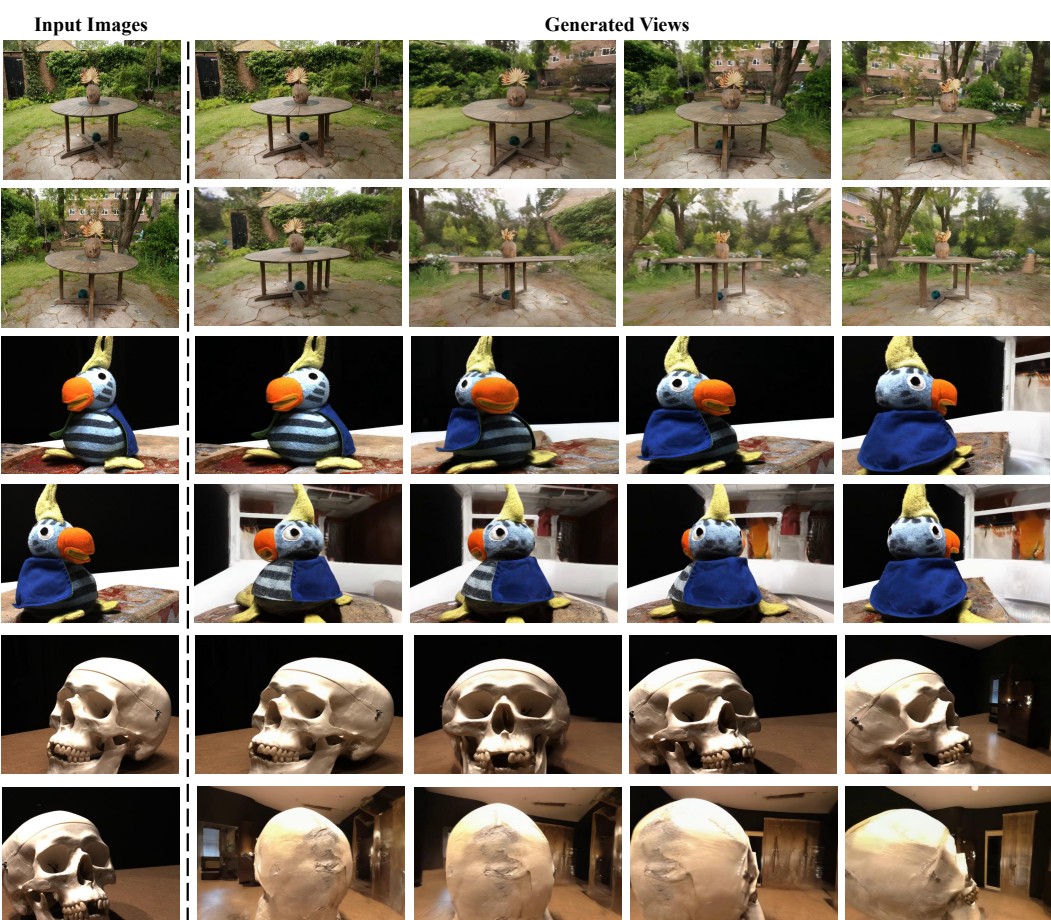

Figure 11: Qualitative results of full 360-degree scenes. This incremental approach demonstrates the effectiveness of our ReconX in reconstructing expansive scenes with only two input views.

Input Views                                              Novel Views

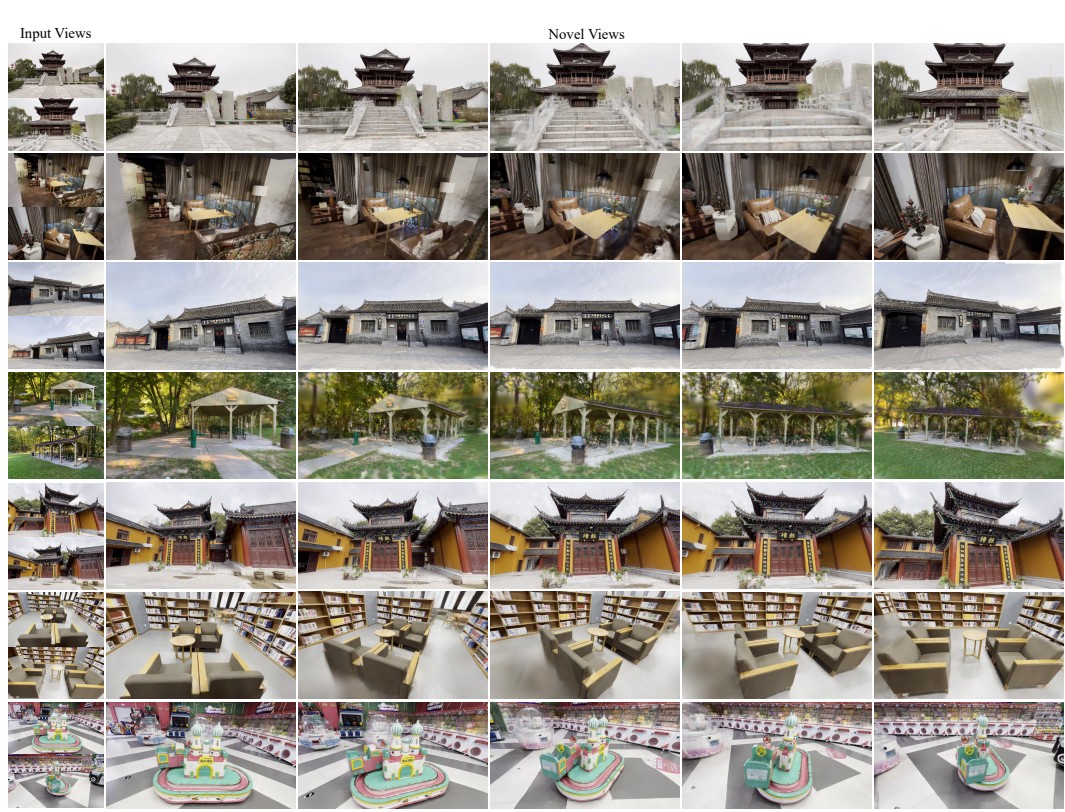

Figure 12: Qualitative results of our ReconX on outdoor scenes Ling et al. (2024).

