# OpenReview forum: "ReconX: Reconstruct Any Scene from Sparse Views with Video Diffusion Model"
_ICLR.cc/2025/Conference — Submitted to ICLR 2025_

### Official Review · Reviewer_JdpZ · 2024-10-19

**Soundness:** 4
**Presentation:** 3
**Contribution:** 3
**Rating:** 6
**Confidence:** 4

**Summary:**

This paper reframes the ambiguous sparse-view reconstruction task as a temporal generation task and unleashes the strong generative prior of large pre-trained video diffusion models. To maintain consistency while generating the video clip, this work proposes to first construct a global point cloud using DUSt3R and then encode it into a contextual space to provide the 3D structure condition. Afterward, it will set the first frame as well as the last frame and repurpose the video diffusion model as an interpolator to synthesize plausible frames. Finally, it will incorporate a confidence-aware 3DGS optimization for reconstruction. Experiments show the paper's superiority in terms of quality and generalization capability to larger viewpoint change compared with previous work.

**Strengths:**

1. The idea of repurposing a video diffusion model as a spatial interpolator sounds reasonable and can provide generative priors to unseen regions.
2. How this work injects structure-aware guidance by using the point cloud from DUSt3R seems critical and can provide insights for future work.
3. This work achieves visually better results than previous methods and seems to generalize better to large viewpoint changes.
4. The paper is well-written and should be easy to follow as long as the code is released.

**Weaknesses:**

1. One strength of directly applying a feed-forward reconstruction like pixelSplat[1] or MVSplat[2] is the inference speed. The proposed method needs to incorporate DUSt3R, a video diffusion model, and a confidence-aware optimization to get the final result. I wonder about the total time consumption during inference compared with baseline models.
2. From the reviewer's perspective, there seem to be multiple solutions given the first frame and the last frame of a video, would it be better to incorporate camera pose information during the training and inference of the video diffusion model? Or on the other side, is the video diffusion model trying to learn a certain type of camera motion due to the limitation of fine-tuning dataset?
3. This paper proposes to utilize a global point cloud as the 3D structure guidance. From the ablation study in Fig.5, the quality seems to be poor without this guidance, showing the importance of incorporating this point cloud. While DUSt3R itself can provide a global point cloud as the reconstruction result, I would like to see a comparison with DUSt3R apart from MVSplat and pixelSplat. Moreover, pixelSplat and MVSplat do not incorporate a global point cloud as input.
4. This work mainly deals with static scene reconstruction, it would be interesting to reconstruct a dynamic scene in future work, which is also mentioned by the authors in the future work part.

[1] pixelSplat: 3D Gaussian Splats from Image Pairs for Scalable Generalizable 3D Reconstruction, CVPR 2024

[2] MVSplat: Efficient 3D Gaussian Splatting from Sparse Multi-View Images, ECCV 2024

[3] DUSt3R: Geometric 3D Vision Made Easy, CVPR 2024

**Questions:**

Please refer to the weakness part. I will be glad to raise my rate if my concerns can be addressed!

---

> ### Author Response · Authors · 2024-11-21
> **Response to Reviewer JdpZ -1**
>
> Thanks so much for your time and positive feedback! To address your concerns, we present the point-to-point responses as follows.
>
> > **Comment 1:** “One strength of directly applying a feed-forward reconstruction like pixelSplat[1] or MVSplat[2] is the inference speed. The proposed method needs to incorporate DUSt3R, a video diffusion model, and a confidence-aware optimization to get the final result. I wonder about the total time consumption during inference compared with baseline models.”
> >
>
> Thanks for your question. Our method’s total inference time is less than 1.5 minutes per-scene: (a) DUSt3R Structure Estimation: < 10s; (b) Video Diffusion: < 20s; (c) confidence-aware optimization: < 1min. based on 3DGS, the final rendering is dramatically faster, achieving approximately 300 FPS with our renderer.
>
> We agree that the inference speed is an advantage of feed-forward based methods. However, they often suffer from severe performance degradation in complex out-of-distribution scenes, large variance angles from input views, and more than two view settings (e.g., 6 views or 9 views). Our key insight is to incorporate generative priors from video diffusion to address the ill-posed sparse-view reconstruction problem. Therefore, it is unavoidable to introduce some additional computation time to achieve superior quality. It is worthy to note that our method is significant faster than per-scene optimization-based method and even recent methods utilizing generative priors (e.g., CAT3D costs 10+ mins per scene). In comparison to baseline models, our method provides a balance between inference time and the quality of the results.
>
> > **Comment 2:** “From the reviewer's perspective, there seem to be multiple solutions given the first frame and the last frame of a video, would it be better to incorporate camera pose information during the training and inference of the video diffusion model? Or on the other side, is the video diffusion model trying to learn a certain type of camera motion due to the limitation of fine-tuning dataset?”
> >
>
> Thanks for your insightful question. Firstly, the large-scale real-world scenes feature complex object distributions and diverse camera trajectories, posing significant challenges when relying on camera poses for conditioning. Meanwhile, mapping camera parameters to corresponding videos in large-scale scenes remains inherently ill-posed, further complicating precise control over camera movements. In contrast, we incorporate the 3D structural condition into video diffusion. On one hand, video interpolation models inherently learn implicit camera motion trajectories during pre-training to ensure smooth transitions between the first and last frames. On the other hand, the 3D structure extracted from DUSt3R provides the video model with a rich 3D spatial context, further enhancing its ability to generate plausible and high-quality camera motion trajectories. Specifically, our 3D structural condition does not explicitly inject camera parameters into the video diffusion model for fine-tuning. Instead, it only provides a feasible region in 3D space according to the sparse-view inputs, leaving the model to implicitly hallucinate a feasible camera motion trajectory using its inherent capabilities, avoiding overfitting to our finetune dataset.

---

> ### Author Response · Authors · 2024-11-21
> **Response to Reviewer JdpZ -2**
>
> > **Comment 3:** “This paper proposes to utilize a global point cloud as the 3D structure guidance. From the ablation study in Fig.5, the quality seems to be poor without this guidance, showing the importance of incorporating this point cloud. While DUSt3R itself can provide a global point cloud as the reconstruction result, I would like to see a comparison with DUSt3R apart from MVSplat and pixelSplat. Moreover, pixelSplat and MVSplat do not incorporate a global point cloud as input.”
> >
>
> Thanks for pointing out this comment. To address your concern, we have revised our paper to add an additional comparison with the point cloud reconstructed only by DUSt3R **in Figure 7 of our revised experiments.** We observe that although the point cloud may not include enough high-quality information, such coarse 3D structure is sufficient to guide the video diffusion in our ReconX to fill in the distortions, occlusions or missing regions. This demonstrates that our ReconX has learned a comprehensive understanding of the 3D scene and can generate high-quality novel views from imperfect conditional information and exhibit robustness to the point cloud conditions.
>
> > **Comment 4:** “This work mainly deals with static scene reconstruction, it would be interesting to reconstruct a dynamic scene in future work, which is also mentioned by the authors in the future work part.”
> >
>
> Thanks for your valuable suggestion! We are also interested in creating a consistent 4D scene from a single image or video. Currently, 4D reconstruction mainly falls into two categories. The first focuses on monocular video-based 4D reconstruction, which may struggle with large-angle novel view rendering due to the limited information available in monocular videos. The second focuses on the multi-view video-based 4D reconstruction, which allows for high-quality rendering within a constrained range of novel views. However, this method typically requires synchronized multi-view videos, which are challenging to obtain in real-world scenarios. Building on these two paradigms, we discover the potential for video diffusion models to explore both directions. For monocular video-based 4D reconstruction, the process begins by obtaining a coarse 4D representation corresponding to the monocular video. This representation serves as an anchor, which can then be warped to novel viewpoints and rendered into new videos. Video diffusion models subsequently refine these outputs and fill in missing or fragmented regions. For the multi-view video-based approach, we leverage the temporal and spatial projection properties of videos. By utilizing video generation models, synchronized multi-view videos can be generated from single images through independent generation and combination along the temporal and spatial dimensions. These consistent synthesized multi-view videos enable the reconstruction of high-quality 4D scenes. We hope that our discussions provide a promising research direction to create consistent 4D scenes with video diffusion models.

---

### Official Review · Reviewer_hNiz · 2024-10-30

**Soundness:** 2
**Presentation:** 2
**Contribution:** 3
**Rating:** 5
**Confidence:** 4

**Summary:**

This paper presents ReconX, which is aim to convert the sparse view 3D reconstruction to the temporal consistency generation by leveraging the video diffusion model as the prior. Moreover, the author uses the 3D structure guidance besides the common multi view images as the guidance, along with the mathematical prove that such new condition can improve the performance. Then, the confidence-aware 3DGS optimization strategy further ensure the performance to have the final novel view synthesis. Comprehensive experiments have been done to demonstrate the in and cross-domain ability of the model.

**Strengths:**

1. Using temporal consistency generation to solving the multi view consistency problem in sparse view reconstruction
2. Including the point cloud as an additional condition for the video generation model and achieving great performance is novel and effective.
3. Applying confidence-aware strategy to optimize the 3DGS to improve the artifacts introduced by the diffusion model.
4. Comprehensive experiments have been done, including different number of views, in and cross-domain, to demonstrate great performance of this work.

**Weaknesses:**

1. More outdoor unbounded dataset can be included such as MipNeRF 360 in the experiment.
2. The performance relies heavily on the point cloud condition. But in the far camera distance case, which is common in MipNeRF 360, the point cloud may not be achieved from the spare input views or the point cloud may not include enough high-quality information, the performance of the diffusion model cannot be preserved.
3. Even though the confidence-aware can boost the performance of 3DGS optimization, but the reason why transformer decoder from  DUSt3R can provide the confidence is not fully investigated while it only draws from empirical study.

**Questions:**

Please refer to the Weakness.

---

> ### Author Response · Authors · 2024-11-21
> **Response to Reviewer hNiz**
>
> We thank the reviewer for valuable feedback. To address your concerns, we present the point-to-point responses as follows.
>
> > **Comment 1: More Outdoor Dataset.** “More outdoor unbounded dataset can be included such as MipNeRF 360 in the experiment.”
> >
>
> We appreciate your constructive comments. Following your suggestions, we show the experiments on MipNeRF-360 **in Figure 4 and Table 3 in our revised paper.** More outdoor datasets like MipNeRF-360 and Tank & Temples with quantitative and qualitative results are also provided our revised paper.
>
> > **Comment 2.** “The performance relies heavily on the point cloud condition. But in the far camera distance case, which is common in MipNeRF 360, the point cloud may not be achieved from the spare input views or the point cloud may not include enough high-quality information, the performance of the diffusion model cannot be preserved.”
> >
>
> Thanks for your valuable comments. We agree that the conditioned point cloud renders may contain artifacts and geometric distortions in the far camera distance case (e.g., MipNeRF-360). Although the point cloud may not include enough high-quality information, such coarse 3D structure is sufficient to guide the video diffusion in our ReconX to fill in the distortions, occlusions or missing regions. **We further verify this by conducting a specific experiment in Figure 7 in our revised paper.** This demonstrates that our ReconX has learned a comprehensive understanding of the 3D scene and can generate high-quality novel views from imperfect conditional information and exhibit robustness to the point cloud conditions.
>
> > **Comment 3.** “Even though the confidence-aware can boost the performance of 3DGS optimization, but the reason why transformer decoder from DUSt3R can provide the confidence is not fully investigated while it only draws from empirical study.”
> >
>
> Thanks for your comment. The training objective of DUSt3R is to map image pairs to 3D space, while the confidence map $\mathcal{C}$ represents the model’s confidence in the pixel matches of image pairs within the 3D scene. Through its training process, DUSt3R inherently assigns low confidence to mismatched regions in image pairs, achieving the goal of Eq. 6. The confidence maps $\left\lbrace\mathcal{C} _ i\right\rbrace _ {i=1}^{K'}$ for each generated frames $\left\lbrace\boldsymbol{I}^i\right\rbrace _ {i=1}^{K'}$  are equivalent to the uncertainty $\sigma_i$. Meanwhile, the pairwise matching between all frames accomplishes the global alignment operation $\mathcal{A}$. **We have revised the method part of our paper to make it clearer.**  If you have additional concerns, we would be pleased to discuss them with you.

---

> ### Author Response · Authors · 2024-11-29
> **Official Comment by Authors**
>
> Dear Reviewer hNiz:
>
> We deeply appreciate your time and effort in reviewing our paper. Your insightful feedback and constructive suggestions have been valuable in improving our work. We have carefully tried to address your concerns with detailed explanations, additional experiments, and a revised version of the manuscript.
>
> With the deadline approaching, we would greatly appreciate it if you could find the time to review our response at your convenience. If there are any remaining questions or unclear explanations, we would be happy to provide further clarification. We sincerely hope that you can consider reevaluating the initial assessment if we have successfully addressed your concerns.
>
> Best Regards,
>
> The Authors

---

> > ### Comment · Reviewer_hNiz · 2024-11-29
> >
> > Thank you for the detailed response. However, according to my concern expressed in the global comment, the **mistake** prevents me from raising the score back to positive.
> > 1. After reviewing the Treehill dataset carefully, I realize even using randomly selected input views, such great results still cannot be reproduced. Because of the complicated camera trajectories in Mip-NeRF 360, it is impossible to select 2 or 3 views with enough overlapping to generate the novel views as the **mistake** version.
> > 2. Moreover, due to the inherent flaw of the 3D decoder in the video diffusion models, the fine details such as the fences in the **mistake** can hardly be achieved without any distortion and degradation.
> >
> > So, I am still worried about the validity of the results and the potential exposure of cross-domain data.

---

> > > ### Author Response · Authors · 2024-11-30
> > > **Response to Reviewer hNiz**
> > >
> > > We still appreciate the detailed explanations and discussions of the reviewer. We would like to clarify that the our mistake is forgetting to update the figure along with the table during rebuttal given the limited time, rather than conducting any experiments incorrectly.
> > >
> > > In the previous version, we only showed Setting-2 in the paper which completely followed the datasplit from ReconFusion and CAT3D. Upon the requests to explain the mistake by the reviewers, we included Setting-1 back to the paper.
> > >
> > > As we designed Setting-1 during rebuttal, we agree that Setting-2 is more proper to be employed for evaluation. Personally I think it would be better to remove Setting-1 from the final version and leave it only as an explanation during the rebuttal period. Though, we believe both experiments well demonstrate the advantages of our method compared with existing ones.

---

### Official Review · Reviewer_4oey · 2024-11-02

**Soundness:** 3
**Presentation:** 2
**Contribution:** 3
**Rating:** 5
**Confidence:** 4

**Summary:**

This paper proposes a framework for obtaining a 3DGS representation of a scene from sparse views, intended for novel view synthesis. It first uses Dust3r to recover a point cloud representation of the scene, which is only a rough estimate of the 3D structure due to the sparsity of views. The point cloud is converted into a learned 3D embedding through an FFN, for injecting as a condition into a video diffusion model to generate intermediate novel views that are consistent with the 3D embedding. Next, each generated view appears to have its pixel values modified by some alignment function, and also assigned a confidence map. The final 3DGS representation is then obtained by standard 3DGS optimization except weighted by the confidence values. Although the proposed method involves per-scene 3DGS optimization, the comparisons are primarily made to feed-forward methods (e.g. MuRF, pixleSplat, MVSplat), and unsurprisingly outperforms them.

**Strengths:**

There does not appear to be previous work combining ideas from Dust3r, video diffusion and 3DGS, so this work is novel in that sense, although there are concurrent ones (ViewCrafter [1], LM-Gaussian [2], 3DGS-Enhancer [3], MVSplat360 [4]) that explore similar ideas.

The proposed methodology appears to be sound, and mostly well-justified. Although this perhaps explains convergent ideas as highlighted by the concurrent works.

Most of the paper is mostly well-written and diagrams are clear, except for a few parts highlighted in the Weaknesses section.

References
- [1] Wangbo Yu, Jinbo Xing, Li Yuan, Wenbo Hu, Xiaoyu Li, Zhipeng Huang, Xiangjun Gao, Tien-Tsin Wong, Ying Shan, Yonghong Tian, ViewCrafter: Taming video diffusion models for high-fidelity novel view synthesis, arXiv preprint arXiv:2409.02048, 3 Sep 2024.
- [2] Hanyang Yu, Xiaoxiao Long, Ping Tan, LM-Gaussian: Boost sparse-view 3D Gaussian splatting with large model priors, arXiv preprint arXiv:2409.03456v1, 5 Sep 2024.
- [3] Xi Liu, Chaoyi Zhou, Siyu Huang, 3DGS-Enhancer: Enhancing unbounded 3D Gaussian splatting with view-consistent 2D diffusion priors, NeurIPS 2024. Also as arXiv preprint arXiv:2410.16266, 21 Oct 2024.
- [4] Yuedong Chen, Chuanxia Zheng, Haofei Xu, Bohan Zhuang, Andrea Vedaldi, Tat-Jen Cham, Jianfei Cai, MVSplat360: Benchmarking 360° generalizable 3D novel view synthesis from sparse views, NeurIPS 2024.

**Weaknesses:**

**Problems with experimental settings**

The primary weakness of this paper lies in the experimental section. The baseline methods chosen seem to be the wrong ones to compare against, as they are all _feed-forward_ methods that produce 3DGS representations (pixelSplat, MVSplat) or radiance fields (MuRF, pixelNeRF, GPNR), done _in a single pass_. In contrast, the proposed method is based on per-scene 3DGS optimization, which unsurprisingly will perform better.

Furthermore, the choice of datasets for testing is limited. For the main results, only RealEstate10k and ACID, with LLFF and DTU only tested on for cross-dataset generalization (which, as an experiment setting itself, is much less impacted in a per-scene optimization method, compared to feed-forward methods). The paper also mentions DL3DV-10K (L373), but it is unclear how and where this is used in the experiments.

Instead, the appropriate baselines, settings and choice of datasets should follow that of per-scene sparse view NVS papers, such as conducted in ReconFusion (Wu 2024b) and Cat3D (Gao 2024), for a fairer comparison.

**Other weaknesses**

A global alignment function is mentioned in L293, but this is neither defined nor is there explanation for its purpose. Based on the brief expression in L293, it appears to do some kind of color adjustment of the video diffusion generated images.

L293-295:
> Through empirical study, we find a well-aligned mapping function $\mathcal{A}$ from the transformer decoder of DUSt3R, which builds the confidence maps ...

This statement is very vague and contradictory. Does $\mathcal{A}$ do color adjustment, or does it produce confidence maps? Is it a handcrafted function? How does it relate to the Dust3r transformer decoder?

The utility of Proposition 1 (L185-189) is unclear. It appears to express that having a 3D condition is less ill-posed than a 2D condition, which is already universally accepted. If so, it would be a distraction from the key ideas of the paper, so I would suggest leaving this out of the paper.

Minor point, but for equation 5 (L256), it would seem that $c_{view}$ and $c_{struc}$ should be deterministically tied to any sampled $x\sim p$. So there is no separate marginalization, and should not appear in the subscript of the expectation $\mathbb{E}$.

Also minor formatting issue, but having Table 4 (L432-436) placed above Table 3 (L468-478) is strange and breaks the flow.

The supplementary video only includes results from the authors' method, but not from the baselines. To better judge qualitative improvements, it would be much better to include video results for all compared methods side-by-side. The few frames in the main paper are not really sufficient, and can easily be cherry-picked.

**Questions:**

Please address the issues raised in the weaknesses section. Specifically:

- If there is time, please provide experimental comparisons with per-scene sparse input NVS methods as elaborated above, in terms of quantitative results, and if possible side-by-side video results.
  - If there are good reasons why it is not appropriate to compare to these baselines, please explain.
- Define the global alignment function $\mathcal{A}$ and explain its role.
- Clarify the vague statement in L293-295, regarding confidence maps for generated images.
- Clarify how the DL3DV-10K dataset is used in the experiments.

REVISED review: Given the major effort the authors have put into the response and revision, I have raised my rating to 5. It is not higher, however, because many of the issues were very obvious (e.g. comparison to feed-forward methods only), and could have been avoided at point of submission. As such, there is insufficient time during the rebuttal period for proper due diligence, on both the part of the authors and reviewers.

---

> ### Author Response · Authors · 2024-11-21
> **Response to Reviewer 4oey -1**
>
> Thanks for your time and comments. To address your concerns, we present the point-to-point response as follows.
>
> > **Comment 1-1: Experimental Settings.** “The primary weakness of this paper lies in the experimental section. The baseline methods chosen seem to be the wrong ones to compare against, as they are all *feed-forward* methods that produce 3DGS representations (pixelSplat, MVSplat) or radiance fields (MuRF, pixelNeRF, GPNR), done *in a single pass*. In contrast, the proposed method is based on per-scene 3DGS optimization, which unsurprisingly will perform better.”
> >
>
> Thanks. Our key insight is to unleash the generative prior of video diffusion by conditioning on 3D structure to compensate for the lack of dense views. Thus we originally intended to compare our method with scene-level sparse-view reconstruction works that incorporate 3D data prior. However, methods like CAT3D and ReconFusion, which focus on sparse-view reconstruction with per-scene optimization, have not been open-sourced, and there were no open-sourced works using video diffusion to incorporate priors before ICLR submission deadline. As a result, we chose to compare our method with feed-forward models like MVSplat and PixelSplat, which train networks with 3D scene data, as these networks inherently learn certain 3D data prior.  In other words, our goal was to demonstrate that using video diffusion as a source of 3D priors allows for better performance in sparse-view reconstruction compared to training a network from scratch with 3D data.
>
> We agree that a comparison with per-scene optimization methods would make the work more comprehensive. Following your suggestions, we have revised our paper to compared both feed-forward based and per-scene optimization based methods. **Specifically, we compare our ReconX with vanilla 3DGS, SparseNeRF, and DNGaussian on more challenging datasets like MipNeRF-360 and the Tank & Tenples datasets, considering different cases with 2,3,6 and 9 views (See Figure 4 and Table 3 in our revision).**
>
> > **Comment 1-2: Dataset Choice.** “Furthermore, the choice of datasets for testing is limited. For the main results, only RealEstate10k and ACID, with LLFF and DTU only tested on for cross-dataset generalization (which, as an experiment setting itself, is much less impacted in a per-scene optimization method, compared to feed-forward methods). The paper also mentions DL3DV-10K (L373), but it is unclear how and where this is used in the experiments.”
> >
>
> Thanks for your comments. To address your concern, we revised our paper to compare our method with recent per-scene optimization-based methods for sparse view reconstruction on more challenging generalizable datasets MipNeRF-360 and Tank & Temples datasets **(See Figure 4 and Table 3 in our revision)**. Regarding the DL3DV dataset, we trained our model on this to demonstrate its performance on outdoor scenes. Due to the limitations of feed-forward methods on this dataset, we did not present quantitative results in the main paper, as these methods fail on it. However, to highlight our model’s strengths in outdoor environments, we have included visual results in the supplementary video and have added comparisons with per-scene optimization methods in **Table 3 from our revision**. We have also provided more visual results on DL3DV in the revised appendix **(See Figure 10)**.
>
> > **Comment 1-3: Baseline Choice.** “Instead, the appropriate baselines, settings and choice of datasets should follow that of per-scene sparse view NVS papers, such as conducted in ReconFusion (Wu 2024b) and Cat3D (Gao 2024), for a fairer comparison.”
> >
>
> Thanks. Following your suggestions, **we have added the qualitative comparisons with ReconFusion and CAT3D in our revision (See Figure 5)**. Since the code and experimental data splits for CAT3D and ReconFusion are not available, we downloaded their results directly from their project pages, using three input views as provided in their papers.

---

> ### Author Response · Authors · 2024-11-21
> **Response to Reviewer 4oey - 2**
>
> > **Comment 2-1 & 2-2: Global Alignment Function.** “A global alignment function is mentioned in L293, but this is neither defined nor is there explanation for its purpose. Based on the brief expression in L293, it appears to do some kind of color adjustment of the video diffusion generated images. Through empirical study, we find a well-aligned mapping function from the transformer decoder of DUSt3R, which builds the confidence maps ...
> >
> >
> > This statement is very vague and contradictory. Does $\mathcal{A}$  do color adjustment, or does it produce confidence maps? Is it a handcrafted function? How does it relate to the Dust3r transformer decoder?
> >
>
> Thanks for your comments. The global align function $\mathcal{A}$ is a tailored global align function to establish connections between each frame and the other frames, enabling a more robust global uncertainty estimation. The insight behind designing this global alignment function is that all video frames essentially correspond to a single 3D scene, with frames influencing one another. Therefore, we propose this global alignment function to establish connections between each frame and the entire video, leading to a more robust global uncertainty estimation.
>
> Specifically, the training objective of DUSt3R is to map image pairs to 3D space, while the confidence map $\mathcal{C}$ represents the model’s confidence in the pixel matches of image pairs within the 3D scene. Through its training process, DUSt3R inherently assigns low confidence to mismatched regions in image pairs, achieving the goal of Eq. 6. The confidence maps $\left\lbrace\mathcal{C} _ i\right\rbrace_{i=1}^{K'}$ for each generated frames $\left\lbrace\boldsymbol{I}^i\right\rbrace_{i=1}^{K'}$  are equivalent to the uncertainty $\sigma_i$. Meanwhile, the pairwise matching between all frames accomplishes the global alignment operation $\mathcal{A}$. **We have revised the method part of our paper to make it clearer.**  If you have additional concerns, we would be pleased to discuss them with you.
>
> > **Comment 2-3: Proposition 1.** “The utility of Proposition 1 (L185-189) is unclear. It appears to express that having a 3D condition is less ill-posed than a 2D condition, which is already universally accepted. If so, it would be a distraction from the key ideas of the paper, so I would suggest leaving this out of the paper.”
> >
>
> Thanks for your comments. We agree with your point and have made efforts to minimize the prominence of Proposition 1 in the main text. To maintain the overall logic flow of the paper, we only keep the insight from Propostion 1 in the motivation section, as we have moved the extensive derivation to the Appendix with more rigorous corrections to the formulation. We hope this revision strikes a better balance while keeping the key ideas of the paper clear and focused.
>
> > **Comment 2-3: Minor Equation issue.** “c_struc and c_view.”
> >
>
> Thanks for pointing out this issue. We have revised this part in our paper, removing the $c_{view}$ and $c_{struc}$ from subscript of the expectation.
>
> > **Comment 2-4: Minor Formatting Issue.** “Also minor formatting issue, but having Table 4 (L432-436) placed above Table 3 (L468-478) is strange and breaks the flow.”
> >
>
> Thanks for pointing out this issue. We have revised the paper to adjust the placement of Table 3 and Table 4 to maintain a smoother flow in our paper.
>
> > **Comment 2-5.** “The supplementary video only includes results from the authors' method, but not from the baselines. To better judge qualitative improvements, it would be much better to include video results for all compared methods side-by-side. The few frames in the main paper are not really sufficient, and can easily be cherry-picked.”
> >
>
> Thanks for your suggestions. **we have revised our paper to add the comparisons in 3D Gaussians with frame-by-frame results in Figure 9.** Following your suggestions, we will provide more comparison video results in demo video if our paper is accepted.
>
> > **Question 1.** “If there is time, please provide experimental comparisons with per-scene sparse input NVS methods as elaborated above, in terms of quantitative results, and if possible side-by-side video results. If there are good reasons why it is not appropriate to compare to these baselines, please explain.”
> >
>
> Thanks for pointing out the question. **We answer the question in above comment (See comment 1-1, 1-2, and 1-3).**
>
> > **Question 2 and 3.** “Define the global alignment function $\mathcal{A}$ and explain its role. Clarify the vague statement in L293-295, regarding confidence maps for generated images.”
> >
>
> Thanks. **We answer the question in above comment (See comment 2-1 and 2-2).**
>
> > **Question 4.** “Clarify how the DL3DV-10K dataset is used in the experiments.”
> >
>
> Thanks. **We answer the question in Comment 1-2 above.**

---

> ### Author Response · Authors · 2024-11-29
> **Official Comment by Authors**
>
> Dear Reviewer 4oey:
>
> We deeply appreciate your time and effort in reviewing our paper. Your insightful feedback and constructive suggestions have been valuable in improving our work. We have carefully tried to address your concerns with detailed explanations, additional experiments, and a revised version of the manuscript.
>
> With the deadline approaching, we would greatly appreciate it if you could find the time to review our response at your convenience. If there are any remaining questions or unclear explanations, we would be happy to provide further clarification. We sincerely hope that you can consider reevaluating the initial assessment if we have successfully addressed your concerns.
>
> Best Regards,
>
> The Authors

---

### Official Review · Reviewer_9tNY · 2024-11-02

**Soundness:** 3
**Presentation:** 3
**Contribution:** 2
**Rating:** 5
**Confidence:** 5

**Summary:**

ReconX proposes video diffusion priors conditioned with 3D structure guidance for reconstruction and novel view synthesis in a 2-view setting. Specifically, the authors leverage the DUSt3R method to obtain a global dense point cloud and encode it into the latent space of the video diffusion model with point transformers for providing sufficient 3D context and obtaining a 3D-consistent novel view coherent with the input images. Finally, a 3D-confidence-aware optimization scheme with 3DGS robustly reconstructs the scene in a few iterations. Experiments show that ReconX outperforms related 2-view baselines - pixelSplat and MVSplat on both in-distribution and OOD benchmarks.

**Strengths:**

1. The proposed 3D point cloud conditioning for ensuring the 3D consistency of generated frames is novel and intuitive.
2. The video diffusion architecture is well-ablated, and each of the highlighted contributions impacts the final reconstruction quality positively.
3. Extensive experiments demonstrate ReconX's ability to achieve high-quality reconstructions that outperform related state-of-the-art methods, particularly in challenging scenarios where there is a large angle variance between input views. ReconX also shows robust generalizability across datasets, a notable limitation of baselines pixelSplat and MVSplat.

**Weaknesses:**

1. Weak Benchmarks: ReconX shows strong sparse-view reconstruction capabilities for scenes from LLFF and DTU datasets, but these are relatively simpler benchmarks, and since the method uses strong video diffusion priors, I would expect a comparison on more challenging benchmarks like MipNeRF360 (or Tanks and Temples). CAT3D / ReconFusion already provides data splits for 3, 6, and 9 view settings for this dataset, so comparisons with a generalized N-view setting would strengthen the submission further.
One can also estimate a DUSt3R point cloud at low cost for N <= 9, so I do not find any bottlenecks in that regard.

2. Weak Baselines: Both baselines - pixelSplat and MVSplat are non-generative in nature. As such, comparison with methods integrating video diffusion priors for sparse-view reconstruction like CAT3D and V3D seems quite relevant.

3. Missing Ablation: The method uses a DUSt3R point cloud initialization for optimizing 3DGS. However, the baseline involving optimization of 3DGS with this dense stereo point cloud is missing from Table 3. Since the authors optimize 3D Gaussians for only 1000 iterations without Adaptive Density Control, my guess is that the DUSt3R point cloud already provides a very strong scene structure. As such, assessing the impact of the video diffusion priors becomes more important. I assume the variant denoted as “base” in Table 3 already integrates the DUSt3R priors with 3DGS.

References:

1. CAT3D: Create Anything in 3D with Multi-View Diffusion Models, Ruiqi Gao* and Aleksander Holynski* and Philipp Henzler and Arthur Brussee and Ricardo Martin-Brualla and Pratul P. Srinivasan and Jonathan T. Barron and Ben Poole*, 2024
2. V3D: Video Diffusion Models are Effective 3D Generators, Zilong Chen and Yikai Wang and Feng Wang and Zhengyi Wang and Huaping Liu, 2024

**Questions:**

1. Since DUSt3R is a pose-free stereo estimation pipeline, the estimated point cloud would not be aligned with the ground truth poses of benchmarks like LLFF/DTU. To obtain synthesized novel views, how is the reconstructed Gaussian point cloud aligned with the COLMAP cameras during inference? I did not find any information regarding that.

2. Does the choice of the video diffusion model have a major impact on the final reconstruction? How would Stable Video Diffusion, for example, work in combination with the proposed 3D structure conditioning? Is there any specific reason behind picking DynamiCrafter?

3. How many scenes from the 3 datasets - RealEstate10k, ACID, and DL3DV-10K are used for training the video diffusion model? I do not see a mention of the total count, so I am assuming all?

4. Can you please provide some more details on the transformer-based encoder for embedding the DUSt3R point cloud? Is it one of the pretrained point Transformer architectures (or maybe the point-SAM encoder), and is it also finetuned along with the video diffusion model? From Figure 2, I am guessing not, but such encoders are usually trained with point clouds from Scannet-like benchmarks, and DUSt3R point clouds would most likely be OOD in terms of overall structure and density.

---

> ### Author Response · Authors · 2024-11-21
> **Response to Reviewer 9tNY - 1**
>
> We sincerely appreciate your constructive and thorough comments. Following your suggestions, we have revised our paper, taking your comments into account. If you have additional concerns, we would be pleased to discuss them with you.
>
> > **Comment 1 & 2: Benchmarks and Baselines.** “ReconX shows strong sparse-view reconstruction capabilities for scenes from LLFF and DTU datasets, but these are relatively simpler benchmarks, and since the method uses strong video diffusion priors, I would expect a comparison on more challenging benchmarks like MipNeRF360 (or Tanks and Temples). CAT3D / ReconFusion already provides data splits for 3, 6, and 9 view settings for this dataset, so comparisons with a generalized N-view setting would strengthen the submission further. One can also estimate a DUSt3R point cloud at low cost for N <= 9, so I do not find any bottlenecks in that regard. Both baselines - pixelSplat and MVSplat are non-generative in nature. As such, comparison with methods integrating video diffusion priors for sparse-view reconstruction like CAT3D and V3D seems quite relevant.”
> >
>
> Thanks for your valuable comments. Following your suggestions, we have conducted experiments in our revised paper **on more challenging benchmarks (i.e., MipNeRF360 and Tanks & Temples) for 2, 3, 6, and 9 view settings following the open-sourced sparse-view reconstruction work like SparseNeRF and DNGaussian (See Figure 4 and Table 3 in our revised version)**. To further demonstrate the quality of our method, we **compare ReconX with CAT3D and ReconFusion qualitatively (See Figure 5) and quantitatively (See Table 5).** Since CAT3D and ReconFusion do not have released codes, we can only obtain the qualitative results by downloading from their project pages with three input views provided in their papers. Qualitative and quantitative comparisons demonstrate the strong sparse-view reconstruction capabilities of our ReconX against various baselines from more challenging datasets.
>
> > **Comment 3: More Ablations.** “The method uses a DUSt3R point cloud initialization for optimizing 3DGS. However, the baseline involving optimization of 3DGS with this dense stereo point cloud is missing from Table 3. Since the authors optimize 3D Gaussians for only 1000 iterations without Adaptive Density Control, my guess is that the DUSt3R point cloud already provides a very strong scene structure. As such, assessing the impact of the video diffusion priors becomes more important. I assume the variant denoted as “base” in Table 3 already integrates the DUSt3R priors with 3DGS.”
> >
>
> Thanks for pointing out this comment. We are not entirely sure if we misunderstood your comment regarding ablation of DUSt3R point cloud initialization. If we have misunderstood, please feel free to point it out so we can further clarify.
>
> To address your concern, we have extended our ablation to **ablate the impact of DUSt3R and video diffusion priors (See Table 4)** respectively and visualize it **in Figure 7 of our revised paper.** Although DUSt3R point cloud already provides a very strong scene structure, the renderings of the point cloud still exhibit occlusions and missing regions. **The second row of Figure 7** displays the corresponding novel views produced by ReconX, showing its ability to fill the missing regions and correct the inaccurate geometry details. This demonstrates that our ReconX has learned a comprehensive understanding of the 3D scene and can generate high-quality novel views from imperfect conditional information and exhibit robustness to the point cloud conditions.
>
> > **Question 1.** “Since DUSt3R is a pose-free stereo estimation pipeline, the estimated point cloud would not be aligned with the ground truth poses of benchmarks like LLFF/DTU. To obtain synthesized novel views, how is the reconstructed Gaussian point cloud aligned with the COLMAP cameras during inference? I did not find any information regarding that.”
> >
>
> Thanks for pointing out the question. It’s true that the camera pose from DUSt3R is not aligned with the COLMAP cameras. Since we adopt the point cloud and camera poses from DUSt3R in our experiment, it is necessary for us to unify the training and testing images into the same DUSt3R coordinate system. Specifically, we process the training and testing images together through DUSt3R to obtain the corresponding camera poses, and utilize only the point maps from the training set as the initial point cloud for optimizing 3DGS. **In the revision, we have added the above details in the Appendix.**

---

> ### Author Response · Authors · 2024-11-21
> **Response to Reviewer 9tNY - 2**
>
> > **Question 2: The Choice of Video Diffusion Model.** “Does the choice of the video diffusion model have a major impact on the final reconstruction? How would Stable Video Diffusion, for example, work in combination with the proposed 3D structure conditioning? Is there any specific reason behind picking DynamiCrafter?”
> >
>
> Thanks. When we propose our work, we carefully considered the future potential of open-sourced contributions and accessibility. While Stable Video Diffusion offers publicly available model weights, its official training code and data processing details have not fully disclosed. In contrast, DynamiCrafter provides full transparency, with both training details and model weights openly available. This commitment to open-source principles was a key factor in our decision to choose DynamiCrafter over other models and our ReconX would be fully open-sourced upon acceptance. Of course, our proposed framework can be seamlessly integrated into Stable Video Diffusion as long as the official training code for Stable Video Diffusion is open-sourced.
>
> > **Question 3.** “How many scenes from the 3 datasets - RealEstate10k, ACID, and DL3DV-10K are used for training the video diffusion model? I do not see a mention of the total count, so I am assuming all?”
> >
>
> Yes. We use all training scenes from RealEstate10K, ACID, and DL3DV-10K.
>
> > **Question 4: More Details.** “Can you please provide some more details on the transformer-based encoder for embedding the DUSt3R point cloud? Is it one of the pretrained point Transformer architectures (or maybe the point-SAM encoder), and is it also finetuned along with the video diffusion model? From Figure 2, I am guessing not, but such encoders are usually trained with point clouds from Scannet-like benchmarks, and DUSt3R point clouds would most likely be OOD in terms of overall structure and density.”
> >
>
> For the transformer-based encoder, we encode the DUSt3R point cloud data to a fixed-length sparse representation of the point cloud. Specifically, we first employ a subsampling based on farthest point sampling (FPS) to reduce the point cloud to a smaller set of key points while retaining its overall structural characteristics. Subsequently, cross-attention is applied between the embeddings of the original point cloud and the downsampled point cloud. This mechanism can be interpreted as a form of "partial" self-attention, where the downsampled points act as query anchors that aggregate information from the original point cloud.
>
> The encoder is not initialized from any pretrained models. Instead, it is trained jointly with the video diffusion model in an end-to-end manner. This design choice ensures that the encoder is specifically adapted to the characteristics of DUSt3R point clouds in our experiment datasets.
>
> **In the revision, we have added the above details in the Appendix.**

---

> ### Author Response · Authors · 2024-11-29
> **Official Comment by Authors**
>
> Dear Reviewer 9tNY:
>
> We deeply appreciate your time and effort in reviewing our paper. Your insightful feedback and constructive suggestions have been valuable in improving our work. We have carefully tried to address your concerns with detailed explanations, additional experiments, and a revised version of the manuscript.
>
> With the deadline approaching, we would greatly appreciate it if you could find the time to review our response at your convenience. If there are any remaining questions or unclear explanations, we would be happy to provide further clarification. We sincerely hope that you can consider reevaluating the initial assessment if we have successfully addressed your concerns.
>
> Best Regards,
>
> The Authors

---

### Official Review · Reviewer_Vp2m · 2024-11-03

**Soundness:** 3
**Presentation:** 3
**Contribution:** 3
**Rating:** 8
**Confidence:** 5

**Summary:**

The paper proposes a method or 3D scene reconstruction from as few as two views via a two-stage approach. First, a finetuned video diffusion model generates multiple view that interpolate the camera trajectory between the given input views. In order to improve 3D consistency of these generated images, the authors introduce a 3D structure conditioning by encoding a point cloud obtained from an existing learned stereo reconstruction method (DUSt3R). In a second stage, the generated views are fused into a 3D representation, for which the authors extend 3D Gaussian Splatting by a weighting based on confidence of DUSt3R.
Experimental evaluation results show strong performance for both in training distribution data as well as cross-dataset generalization, outperforming state-of-the-art baselines. Ablation studies validate the effectiveness of the technical contributions.

**Strengths:**

- The paper tackles an important and very challenging task of reconstructing general scenes from very sparse views.
- The two-stage approach leverages the strong prior of video diffusion models and the technical contributions fit well into this framework.
- The experimental evaluation shows strong results consistently outperforming state-of-the-art baselines on both
   - in training distribution data
     - significant quantitative and qualitative advantage for small angle variance in input views (table 1, figure 3)
     - even larger gap for large angle variance in input views (table 2, figure 4)
  - and cross-dataset generalization
    - baselines fail completely, while the proposed method achieves reasonable metric scores (table 4)
    - impressive qualitative results (figure 6)
- The ablation study validates the effectiveness of the proposed 3D structure conditioning and the confidence-aware 3DGS optimization (especially convincing qualitative ablation in figure 5).
- The paper is mostly well-written and easy to follow.
- The supplemental video provides additional convincing qualitative reconstruction results.

**Weaknesses:**

- The lack of precise mathematical formulation raises doubt about proposition 1 and its proof:
  - The main inequality in equation 17 (appendix) is justified only verbally and is not obvious to me.
  - A counterexample for the inequality in line 841 could be a dataset mainly consisting of dark rooms with all ground truth renderings being black except for one where the lights are on (with possibly very complex geometry or even transparent and reflective materials). For this case, fitting the marginal distribution can be easier than fitting the conditional distribution, if the given scene s happens to be the one with lights on.
  - Moreover, if the 3D structure (point cloud) is obtained from 2D images only via stereo reconstruction (DUSt3R [1]), then an image encoder should be theoretically able to encode the images in the same way as the composition of DUSt3R and the point cloud encoder used in the method. Following this argument, a strict inequality as in proposition 1 is not reasonable.

- Some lack of clarity:
  - What exactly is the implementation of the learnable embedding function PosEmb in equation 3?
  - What is meant with "uncertainty of unconstrained images" (line 265)? Does this mean that generated images are not fully 3D-consistent?
  - Regarding the confidence-aware 3DGS optimization in section 4.4, it is unclear how the loss in equation 6 is still used / relevant for the final loss in equation 7, which does not use equation 6 anymore.

- Somewhat unfair comparison with baselines:
  - The proposed method relies heavily on the strong priors learned by DUSt3R [1] and the video diffusion model.
  - All baselines are trained more or less from scratch on small-scale datasets compared to the large-scale datasets for training video diffusion models.
  - Therefore, it is not that surprising that the proposed method generalizes much better to other datasets out of the training/finetuning data distribution.

- Restriction to view interpolation:
  - Given that the method makes use of a pre-trained video diffusion model, the restriction to interpolation of the camera trajectory between two input views is unsatisfactory.
  - In many cases, two views as conditioning already eliminate uncertainty in the reconstruction task entirely such that deterministic approaches like pixelSplat [4] and MVSplat [5] already produce detailed and sharp reconstructions.
  - The proposed generative approach with a strong prior trained on large-scale single view data should be able to extrapolate from input views to some extent, which the paper misses to evaluate.
  - This is especially critical, if the paper motivates the method with the limitation of generalizable 3D reconstruction methods that "struggle to generate high-quality images in areas not visible from the input perspectives" (lines 143f.).

- Missing relevant related work and possible baseline:
  - latentSplat [2] aims to bridge the gap between regression models for generalizable NVS and generative models for 3D reconstruction and is therefore a very relevant work and potential baseline.
  - GeNVS [3] was one of the first approaches to generative novel views with a diffusion model conditioned on 3D-aware (pixelNeRF) features and is therefore important related work.

Minor comments:
- The paper sometimes uses the terms "conditioning" and "guidance" interchangeably, although they have different meanings to me (conditioning: giving something as additional input to the diffusion model; guidance: using CFG or also gradients (e.g. classifier guidance) to affect the sampling process). A more precise use of these terms would be helpful.
- The ablation study in table 3 could be extended to contain all different combinations of leaving out individual components in order to find possible dependencies between them.

References:
- [1] DUSt3R: Geometric 3D Vision Made Easy. CVPR 2024
- [2] latentSplat: Autoencoding Variational Gaussians for Fast Generalizable 3D Reconstruction. ECCV 2024
- [3] Generative Novel View Synthesis with 3D-Aware Diffusion Models. ICCV 2023
- [4] pixelSplat: 3D Gaussian Splats from Image Pairs for Scalable Generalizable 3D Reconstruction. CVPR 2024
- [5] MVSplat: Efficient 3D Gaussian Splatting from Sparse Multi-View Images. ECCV 2024

**Questions:**

- Is the mentioned counterexample for the proof correct or am I missing something?
- What exactly is the implementation of the learnable embedding function PosEmb in equation 3?
- What is meant with "uncertainty of unconstrained images" (line 265)? Does this mean that generated images are not fully 3D-consistent?
- Regarding the confidence-aware 3DGS optimization in section 4.4, how is the loss in equation 6 still used / relevant for the final loss in equation 7, which does not use equation 6 anymore?
- Is there any specific reason why you limit yourself to the case of view interpolation and do not evaluate extrapolation?

---

> ### Author Response · Authors · 2024-11-21
> **Response to Reviewer Vp2m - 1**
>
> We sincerely thank you for your time and valuable comments. To address your concerns, we present the point-to-point response as follows.
>
> > **Comment 1-1 & Question 1: Mathematical Formulation.** “The main inequality in equation 17 (appendix) is justified only verbally and is not obvious to me. A counterexample for the inequality in line 841 could be a dataset mainly consisting of dark rooms with all ground truth renderings being black except for one where the lights are on (with possibly very complex geometry or even transparent and reflective materials). For this case, fitting the marginal distribution can be easier than fitting the conditional distribution, if the given scene s happens to be the one with lights on.”
> >
>
> Thanks for your insightful question. We are not entirely sure if we misunderstood your comment regarding mathematical formulation. If we have misunderstood, please feel free to point it out so we can further clarify.
>
> We agree your point in cases where data distribution is extremely imbalanced. However, such imbalanced cases are rare in general 3D scene datasets, thus our assumption (i.e., fitting the 3D conditional distribution is simpler) is more manageable). Motivated by your counterexample and to ensure the robustness of our formulation, we have made the following modifications to address your concern in a more rigorous manner.
>
> We have revised the inequality $\min_E \mathcal{D}\left(q(\boldsymbol{x} \mid s) \| p_{\boldsymbol{\theta}, g, E}(\boldsymbol{x})\right)<\min_E \mathcal{D}\left(q(\boldsymbol{x}) \| p_{\boldsymbol{\theta}, g, E}(\boldsymbol{x})\right)$ to hold almost everywhere (a.e.), **in mathematical terms in our revised appendix**, this is defined as: $$\mathcal{P} _ {q(s)} \left\lbrace \min_E \mathcal{D}\left(q(\boldsymbol{x} \mid s) \| p_{\boldsymbol{\theta}, g, E}(\boldsymbol{x})\right)<\min_E \mathcal{D}\left(q(\boldsymbol{x}) \| p_{\boldsymbol{\theta}, g, E}(\boldsymbol{x})\right) \right\rbrace= 1.$$ In the example you provided regarding the dataset composed of "dark room" scenarios, since there is only one scenario where the lights are on, the distribution of scenarios $q(s)$  in the dataset is almost everywhere (a.e.) dominated by the lights-off condition, which means $\mathcal{P}_{q(s)}(s=\text{lights are on})=0$, therefore, the inequality mentioned above holds almost everywhere. In the example your provided, we can rewrite Equation 17 in our paper as:
>
> $$
> \min_\theta E_{q(s)} \min_{g, E} D\left(q(x \mid s) \| p_{\theta, g, E}(x)\right)
> $$
> $$\le \min_{\theta,g} E _ {q(s)} \min_E D\left(q(x \mid s) \| p_{\theta, g,E}(x)\right) = \min_{\theta,g}\int_s q(s)\min_E D\left(q(x \mid s) \| p_{\theta, g,E}(x)\right) ds$$
> $$
> =\min_{\theta,g}\left(\underset{s=\text{light on}}\int q(s)\min_E D\left(q(x \mid s) \| p_{\theta, g,E}(x)\right) ds+\underset{s=\text{light off}}\int q(s)\min_E D\left(q(x \mid s) \| p_{\theta, g,E}(x)\right) ds \right)
> $$
> $$
> =0 + \min_{\theta,g}\underset{s=\text{light off}}\int q(s)\min_E D\left(q(x \mid s) \| p_{\theta, g,E}(x)\right) ds
> $$
> $$
> <\min_{\theta,g}\underset{s=\text{light off}}\int q(s)\min_E D\left(q(x) \| p_{\theta, g,E}(x)\right) ds
> $$
> $$
> =\min_{\boldsymbol{\theta}, g, E} E_{q(s)} {D}\left(q(\boldsymbol{x}) \| p_{\boldsymbol{\theta}, g, E}(\boldsymbol{x})\right)
> =\min_{\boldsymbol{\theta}, g, E} {D}\left(q(\boldsymbol{x}) \| p_{\boldsymbol{\theta}, g, E}(\boldsymbol{x})\right)
> $$
>
> > **Comment 1-2: Mathematical Formulation.** “Moreover, if the 3D structure (point cloud) is obtained from 2D images only via stereo reconstruction (DUSt3R [1]), then an image encoder should be theoretically able to encode the images in the same way as the composition of DUSt3R and the point cloud encoder used in the method. Following this argument, a strict inequality as in proposition 1 is not reasonable.”
> >
>
> Thanks for pointing out this comment. DUSt3R is an end-to-end stereo reconstruction framework trained on large-scale 3D scene datasets, thus encoding rich 3D priors in the encoders beyond 2D images. Previous works like DynamiCrafter and CAT3D usually inject the image condition into diffusion model through the image encoder (e.g., CLIP encoder or VAE encoder) only trained from 2D image dataset, which only contains 2D prior. In contrast, we utilize the 3D prior embedded in the DUSt3R model as the structural guidance, making video diffusion understand the 3D scene more comprehensively.

---

> ### Author Response · Authors · 2024-11-21
> **Response to Reviewer Vp2m - 2**
>
> > **Comment 2-1 & Question 2: Clarity of PosEmb.** “What exactly is the implementation of the learnable embedding function PosEmb in equation 3?”
> >
>
> The PosEmb implemented in our paper is a column-wise positional embedding function: $\mathbb{R}^3 \rightarrow \mathbb{R}^C$, where $C$ is the dimension of embedding. More specifically, the **PosEmb** function is implemented as follows:
>
> 1. **Fixed Sinusoidal Basis**: The basis $\mathbf{e}$ is a 3D sinusoidal encoding: $\mathbf{e} = [\sin(2^0 \pi p), \sin(2^1 \pi p), \dots]$ , where $p \in \mathbb{R}^3$ is the position.
> 2. **Embedding Calculation**: The input $\mathbf{x}$ is projected onto $\mathbf{e}$ and its sine and cosine are concatenated:
>
>      $\mathbf{embeddings} = \text{concat}(\sin(\mathbf{proj}), \cos(\mathbf{proj}))$
>
> 3. **Learnable Transformation**: The positional encoding is passed through an MLP along with the input $\mathbf{x}$: $\mathbf{y} = \text{MLP}(\text{concat}(\mathbf{embeddings}, \mathbf{x}))$
>
> In short, **PosEmb** combines a fixed sinusoidal encoding with a learnable MLP transformation. To address your concern, **we have added the above description in our revised appendix.**
>
> > **Comment 2-2 & Question 3: Clarity of Uncertainty.** “What is meant with "uncertainty of unconstrained images" (line 265)? Does this mean that generated images are not fully 3D-consistent?”
> >
>
> Thanks for pointing out the question. The “uncertainty of unconstrained images” is to measure the minor inconsistency between generated video frames, particularly for the high-frequency details. The generated images cannot guarantee full 3D-consistent. Currently, video diffusion model sometimes fails to deal with the high-frequency details. On one hand, learning the distribution of high-frequency details is challenging for the probabilistic modeling process. On the other hand, VAEs fails to maintain precise high-frequency details in generated videos. This finding is also mentioned by CAT3D. Therefore, it is necessary for us to measure the uncertainty of unconstrained video frames and adopt our proposed confidence-aware 3DGS optimization scheme to achieve a better reconstruction result.
>
> > **Comment 2-3 & Question 4: Clarity of Equation 6.** “Regarding the confidence-aware 3DGS optimization in section 4.4, it is unclear how the loss in equation 6 is still used relevant for the final loss in equation 7, which does not use equation 6 anymore.”
> >
>
> Thanks for your comments. In our formulation, the uncertainty $\sigma$ is estimated by a prediction function (can be implemented as a network) optimized with the loss function in the Equation 6.
>
> The global align function $\mathcal{A}$ is a designed (handcrafted) function to connect each frame with the other frames. The insight behind designing this global alignment function is that all video frames essentially correspond to a single 3D scene, with frames influencing one another. Therefore, we propose this global alignment function to establish connections between each frame and the entire video, leading to a more robust global uncertainty estimation.
>
> In DUSt3R, the confidence map provides a prediction score for each pixel, representing the level of confidence the transformer network assigns to that pixel. Specifically, DUSt3R builds image pairs among our generated video frames, and for each frame, it is coupled with the rest frames to obtain the confidence map. Generally speaking, if there are inconsistencies between the generated frames, it becomes difficult to align in 3D space after being input into Dust3r, resulting in lower confidence scores for these inconsistent regions.
>
> The image pairing among video frames in DUSt3R acts as the role of the global align function $\mathcal{A}$, the confidence map $\mathcal{C}$ from DUSt3R  is used to measure the uncertainty $\sigma$. **Moreover, we have revised our method part following above descriptions to make it clearer.** If you still have concerns, please feel free to point it out so we can further clarify.

---

> ### Author Response · Authors · 2024-11-21
> **Response to Reviewer Vp2m - 3**
>
> > **Comment 3: Comparison with baselines.** “The proposed method relies heavily on the strong priors learned by DUSt3R [1] and the video diffusion model. All baselines are trained more or less from scratch on small-scale datasets compared to the large-scale datasets for training video diffusion models. Therefore, it is not that surprising that the proposed method generalizes much better to other datasets out of the training/finetuning data distribution.”
> >
>
> We appreciate the reviewer bringing this comment up. We answer this question from two perspectives:
>
> - **Why we choose methods like MVSplat and PixelSplat:** In this paper, our key insight and contribution is to unleash the generative prior of video diffusion by conditioning on 3D structure to address the ill-posed sparse-view scene reconstruction problem. However, there were no open-sourced works using video diffusion to incorporate priors before ICLR submission deadline. As a result, we chose to compare our method with methods with 3D data priors like MVSplat and PixelSplat, which train networks with 3D scenes, thus inherently learning certain 3D data prior. In other words, we aim to demonstrate that using video diffusion as an engine of generative 3D priors allows for better performance in sparse-view reconstruction compared to training a deterministic network from scratch with 3D data.
> - **Further comparisons:** To further demonstrate our generalizability and superiority, **we have added qualitative comparisons with CAT3D and ReconFusion in the revised paper (See Figure 5)**. Since CAT3D and ReconFusion do not have released the codes, we can only obtain the qualitative results by downloading from their project pages with three input views provided in their papers. From the results, we observe that our method achieves better rendering details compared with CAT3D and ReconFusion. Moreover, following your suggestions, we have compared with the open-sourced LatentSplat into our feed-forward based baselines quantitatively as below. However, due to the time limitation in rebuttal, as LatentSplat have not trained on ACID dataset and other outdoor datasets like MipNeRF-360, we will train it on various data to give a more comprehensive comparison in final version if our paper is accepted.
>
> **Table 1: Quantitative comparison with LatentSplat on Real10K**
>
> | Small Angle Variance | PNSR | SSIM | LPIPS |
> | --- | --- | --- | --- |
> | LatentSplat | 25.92 | 0.824 | 0.133 |
> | ReconX (Ours) | 28.31 | 0.912 | 0.088 |
> | **Large Angle Variance** |  |  |  |
> | LatentSplat | 20.53 | 0.745 | 0.193 |
> | ReconX (Ours) | 23.70 | 0.867 | 0.143 |
>
> > **Comment 4-1: The Choice of View Interpolation.** “Given that the method makes use of a pre-trained video diffusion model, the restriction to interpolation of the camera trajectory between two input views is unsatisfactory. In many cases, two views as conditioning already eliminate uncertainty in the reconstruction task entirely such that deterministic approaches like pixelSplat [4] and MVSplat [5] already produce detailed and sharp reconstructions.”
> >
>
> Thanks for your insightful question. We agree that two views as conditioning already eliminate some uncertainty in the reconstruction task. However, **when there exists unseen areas between the input views (i.e., large view variance between input images), it is difficult for such deterministic approaches to generate plausible details (See Figure 3 and Table 2 for large angle variance in our revised paper).** In contrast, we utilize the generative prior to extrapolate such areas via 3D structure-guided video diffusion process. Moreover, this two-view geometry serves as the fundamental building block for multi-view recovery, which can be extended to more sparse views (See Table 3 across different numbers of input views in revised paper).

---

> ### Author Response · Authors · 2024-11-21
> **Response to Reviewer Vp2m - 4**
>
> > **Comment 4-2 & Question 5: The Capability of Extrapolation.** “The proposed generative approach with a strong prior trained on large-scale single view data should be able to extrapolate from input views to some extent, which the paper misses to evaluate. This is especially critical, if the paper motivates the method with the limitation of generalizable 3D reconstruction methods that "struggle to generate high-quality images in areas not visible from the input perspectives" (lines 143f.).”
> >
>
> We value your insightful comments. As we use a pair of input views in our method, it is worthy to note that if the angular difference between the two views is too large, it is hard to ensure that the entire interpolated region falls within the visible perspective of the input views, which requires the extrapolation ability. **We have evaluated it in our generalizable experiments with DTU dataset. For instance (See Cross Set experiments in Figure 3)**, we cannot see the roof area from the input views, while our ReconX is able to extrapolate and generate the red and yellow roof with 3D structure-guided generative prior. To further demonstrate the extrapolation capability of our method, **we have added a specific experiment in the revised paper (See Figure 7).** This experiment selects two views with large angular span and highlights the extrapolated regions in the red boxes in the novel rendered views. This emphasizes our model's generative power to extrapolate unseen regions and extend beyond the visible input views.
>
> > **Comment 5: More Relevant Related Work.** "latentSplat [2] aims to bridge the gap between regression models for generalizable NVS and generative models for 3D reconstruction and is therefore a very relevant work and potential baseline. GeNVS [3] was one of the first approaches to generative novel views with a diffusion model conditioned on 3D-aware (pixelNeRF) features and is therefore important related work.”
> >
>
> Thanks so much for bringing these work to us. Follow your valuable suggestions, **we have added LatentSplat and GeNVS into the related work of our revised paper.** Moreover, we compare our method quantitatively with LatentSplat in our revised paper. We believe these revision makes our paper more comprehensive.
>
> > **Comment 6-1:** “The paper sometimes uses the terms "conditioning" and "guidance" interchangeably, although they have different meanings to me (conditioning: giving something as additional input to the diffusion model; guidance: using CFG or also gradients (e.g. classifier guidance) to affect the sampling process). A more precise use of these terms would be helpful.”
> >
>
> We appreciate the reviewer bringing this comment up, and we concur that the use of these terms need to be made clearer. Following your suggestions, we have revised our paper for more precise use of “conditioning” and “guidance”. Specifically, we have changed the title of subsection 4.2 “Building the 3D Structure Guidance” into “Building the 3D Structure Condtiion” with all related description in our revised paper.
>
> > **Comment 6-2:** “The ablation study in table 3 could be extended to contain all different combinations of leaving out individual components in order to find possible dependencies between them.”
> >
>
> Thanks for your valuable comments. Following your suggestions, we have extended our ablation study to contain all different combinations of leaving out individual components in our revised paper **(See Table 4).**

---

> > ### Comment · Reviewer_Vp2m · 2024-11-25
> >
> > Thank you very much for the detailed rebuttal. Most of my concerns are addressed. However, my concern regarding the limitation to target views rather close to the input views remains. I agree that for some cases, the generative prior is effectively leveraged due to occlusions. Still, the restriction to view interpolation only considers the case given strong conditioning. The additional qualitative results in figure 8 indicate some extrapolation capabilities.
> > Since previous works like ReconFusion and CAT3D already leverage incremental / autoregressive approaches in order to reconstruct full 360° scenes, this limitation in the evaluation seems like a step back. Therefore, it would be interesting to see, whether a similar autoregressive approach is applicable in combination with the proposed method.

---

> > > ### Author Response · Authors · 2024-11-25
> > > **Response to Reviewer Vp2m**
> > >
> > > Thanks for your thoughtful feedback! We will try to address your concern in the General Response. Any results or progress on this point will be updated in the General Response before the rebuttal deadline. Thanks again for your valuable suggestion, which has been helpful in guiding our improvements.

---

### Author Response · Authors · 2024-11-21
**General response to reviewers**

We thank all the reviewers for their insightful comments and suggestion! We are glad to see that the reviewers consider our work to be addressing “an important and very challenging task” describing it as “novel”, “well-written”, and “comprehensive”. Their valuable suggestions helped us improve our submission and better strengthen our claims. Here we summarize the major updates brought to the revised manuscript:

1. [Related Work]: We add latentSplat [1] and GeNVS [2] to our related work section
2. [Methodology]: We refine our description about the confidence-aware 3DGS and some minor changes.
3. [Experiment]: We add additional expeiments with more baseline methods on complex outdoor datasets, and refine the ablation study.
4. [Appendix]: We present more experiement details, including the PosEmb, transformer-based encoder, and more evaluation results

Please let us know if you have any question, and we look forward to engaging in more discussions with reviewers to further improve our work.

[1] latentSplat: Autoencoding Variational Gaussians for Fast Generalizable 3D Reconstruction. ECCV 2024

[2] Generative Novel View Synthesis with 3D-Aware Diffusion Models. ICCV 2023

# Quantitative comparison with per-scene optimization based methods on more challenging outdoor datasets

| Dataset | Method | PSNR↑ | SSIM↑ | LPIPS↓ | PSNR↑ | SSIM↑ | LPIPS↓ | PSNR↑ | SSIM↑ | LPIPS↓ | PSNR↑ | SSIM↑ | LPIPS↓ |
| --- | --- | --- | --- | --- | --- | --- | --- | --- | --- | --- | --- | --- | --- |
|  |  |  | 2-view |  |  | 3-view |  |  | 6-view |  |  | 9-view |  |
| **Mip-NeRF 360** | 3DGS | 10.36 | 0.108 | 0.776 | 10.86 | 0.126 | 0.695 | 12.48 | 0.180 | 0.654 | 13.10 | 0.191 | 0.622 |
|  | SparseNeRF | 11.47 | 0.190 | 0.716 | 11.67 | 0.197 | 0.718 | 14.79 | 0.150 | 0.662 | 14.90 | 0.156 | 0.656 |
|  | DNGaussian | 10.81 | 0.133 | 0.727 | 11.13 | 0.153 | 0.711 | 12.20 | 0.218 | 0.688 | 13.01 | 0.246 | 0.678 |
|  | **ReconX (Ours)** | **13.37** | **0.283** | **0.550** | **16.66** | **0.408** | **0.427** | **18.72** | **0.451** | **0.390** | **18.17** | **0.446** | **0.382** |
| **Tank and Temples** | 3DGS | 9.57 | 0.108 | 0.779 | 10.15 | 0.118 | 0.763 | 11.48 | 0.204 | 0.685 | 12.50 | 0.202 | 0.669 |
|  | SparseNeRF | 9.23 | 0.191 | 0.632 | 9.55 | 0.216 | 0.633 | 12.24 | 0.274 | 0.615 | 12.74 | 0.294 | 0.608 |
|  | DNGaussian | 10.23 | 0.156 | 0.643 | 11.25 | 0.204 | 0.584 | 12.92 | 0.231 | 0.535 | 13.01 | 0.256 | 0.520 |
|  | **ReconX (Ours)** | **14.28** | **0.394** | **0.564** | **15.38** | **0.437** | **0.483** | **16.27** | **0.497** | **0.420** | **18.38** | **0.556** | **0.355** |
| **DL3DV** | 3DGS | 9.46 | 0.125 | 0.732 | 10.97 | 0.248 | 0.567 | 13.34 | 0.332 | 0.498 | 14.99 | 0.403 | 0.446 |
|  | SparseNeRF | 9.14 | 0.137 | 0.793 | 10.89 | 0.214 | 0.593 | 12.15 | 0.234 | 0.577 | 12.89 | 0.242 | 0.576 |
|  | DNGaussian | 10.10 | 0.149 | 0.523 | 11.10 | 0.274 | 0.577 | 12.65 | 0.330 | 0.548 | 13.46 | 0.367 | 0.541 |
|  | **ReconX (Ours)** | **13.60** | **0.307** | **0.554** | **14.97** | **0.419** | **0.444** | **17.45** | **0.476** | **0.426** | **18.59** | **0.584** | **0.386** |

---

> ### Comment · Reviewer_4oey · 2024-11-22
>
> Thanks to the authors for a major rework of the experiments. I'm willing to increase my score, and will consider the amount to do so.
>
> However, I feel the baselines chosen are too weak, and gives the wrong sense that ReconX has a large jump over SOTA. Even if Cat3D and ReconFusion are not available, the published quantitative results can still be used. In fact, there seems to be an overlapped experimental setting with Mip-NeRF 360 dataset, and judging by the numbers in [Cat3D](https://openreview.net/pdf?id=TFZlFRl9Ks) Table 1, ReconX is generally still better. But at least the baselines (including Zip-NeRF, ZeroNVS and ReconFusion) are more competitive than the current ones in the table above. Why not use those baselines? I think they would increase the solidity of the paper.

---

> > ### Author Response · Authors · 2024-11-22
> > **Response to Reviewer 4oey**
> >
> > Thanks so much for your time and positive feedback! Since CAT3D and ReconFusion do not have released codes and their specific experimental data split, we can only obtain the qualitative results by downloading from their project pages with three input views provided in their papers. **We compare with CAT3D and ReconFusion qualitatively in our revised paper (See Figure 5).**
> >
> > For quantitative comparison, we agree that adding the quantitative results from published papers of ReconFusion and CAT3D would increase the solidity of our paper. **However, our main concern lies in the selection of input views for evaluation.** Specifically, the **description in ReconFusion and CAT3D** related to input view selection is:
> >
> > - “*For the mip-NeRF 360 dataset we design a **heuristic** to choose a train split of views that are uniformly distributed around the hemisphere and pointed toward the central object of interest: We randomly **sample $10^6$ different 9-view splits** and **use the one that minimizes these heuristic losses**, then further choose the 6- and 3-view splits to be subsets of the 9-view split.*”
> > - “*For the mip-NeRF 360 dataset, we retain its original test set and select the input views from the training set **using a heuristic to encourage reasonable camera spacing and coverage of the central object.***”
> >
> > However, we lack more details of their **meticulously designed heuristic loss with their $10^6$ different 9-view splits, that encourage more reasonable camera spacing and coverage of central object.** Therefore, without any cherry picking, we only choose one random 9-view split in MipNeRF 360 that follows *uniformly distributed around the hemisphere and pointed toward the central object of interest with 6- and 3-view splits to be subsets of the 9-view split.* **Without specific heuristic loss design, our generalizable test setting in MipNeRF 360 is more challenging than CAT3D and ReconFusion.**
> >
> > Despite the bias of our setting in input-view selection with Table-1 in CAT3D, we follow your suggestions to compare them (we add the * to note the difference) as below. Although our test setting is more challenging, our ReconX is generally still better than all baseline methods. **Moreover, we will add the comparisons with above detailed illustration in our final version if our paper is accepted.** Thanks again for bringing up this comment. If you have additional concerns, we would be pleased to discuss them with you.
> >
> > | Dataset | Method | PSNR↑ | SSIM↑ | LPIPS↓ | PSNR↑ | SSIM↑ | LPIPS↓ | PSNR↑ | SSIM↑ | LPIPS↓ |
> > | --- | --- | --- | --- | --- | --- | --- | --- | --- | --- | --- |
> > |  |  |  | 3-view |  |  | 6-view |  |  | 9-view |  |
> > | **Mip-NeRF 360** | 3DGS | 10.86 | 0.126 | 0.695 | 12.48 | 0.180 | 0.654 | 13.10 | 0.191 | 0.622 |
> > |  | SparseNeRF | 11.67 | 0.197 | 0.718 | 14.79 | 0.150 | 0.662 | 14.90 | 0.156 | 0.656 |
> > |  | DNGaussian | 11.13 | 0.153 | 0.711 | 12.20 | 0.218 | 0.688 | 13.01 | 0.246 | 0.678 |
> > |  | Zip-NeRF* | 12.77 | 0.271 | 0.705 | 13.61 | 0.284 | 0.663 | 14.30 | 0.312 | 0.633 |
> > |  | ZeroNVS* | 14.44 | 0.316 | 0.680 | 15.51 | 0.337 | 0.663 | 15.99 | 0.350 | 0.655 |
> > |  | ReconFusion* | 15.50 | 0.358 | 0.585 | 16.93 | 0.401 | 0.544 | 18.19 | 0.432 | 0.511 |
> > |  | CAT3D* | 16.62 | 0.377 | 0.515 | 17.72 | 0.425 | 0.482 | **18.67** | **0.460** | 0.460 |
> > |  | **ReconX (Ours)** | **16.66** | **0.408** | **0.427** | **18.72** | **0.451** | **0.390** | 18.17 | 0.446 | **0.382** |

---

> > > ### Comment · Reviewer_9tNY · 2024-11-22
> > > **ReconFusion / CAT3D splits publicly available**
> > >
> > > Hi,
> > >
> > > I am not sure why you are not able to find the official train/test splits of MipNeRF360 online, as they are clearly linked on the project page of ReconFusion. Please find them here - https://drive.google.com/drive/folders/10oT2_OQ9Sjh5wlfJQoGx2y7ZKYwpgNg5?usp=sharing. I would request you to kindly update your comments and the paper accordingly.

---

> > > > ### Author Response · Authors · 2024-11-22
> > > > **Response to Reviewer 9tNY**
> > > >
> > > > Thanks for reminding us of this significant open-sourced data. We are now running the experiments of ReconX in their open-sourced data and will provide the comparison with ZipNeRF, ZeroNVS, ReconFusion, and CAT3D as soon as possible. If you have more concerns, please feel free to point it out so we can further clarify.

---

> > > > ### Author Response · Authors · 2024-11-22
> > > > **Response to Reviewer 4oey and Reviewer 9tNY**
> > > >
> > > > Thanks for your constructive suggestions and kind reminders! We have followed the open-sourced data split in ReconFusion to conduct an additional experiment in quantitative comparison with Zip-NeRF, ZeroNVS, ReconFusion and CAT3D in the challenging outdoor datasets MipNeRF 360. Moreover, following your suggestions, we have revised our paper to include this comparisons in Table 5. As the input views are filtered by the specific heuristic loss design in CAT3D and ReconFusion, they cover more central of the object in the scene, which makes the test setting easier. We observe that our ReconX demonstrates superior performance than all baseline methods. Thanks again for your time and valuable comments. We believe it would definitely increase the solidity of the paper. If you have additional concerns, we would be pleased to discuss them with you.
> > > >
> > > > | Dataset | Method | PSNR↑ | SSIM↑ | LPIPS↓ | PSNR↑ | SSIM↑ | LPIPS↓ | PSNR↑ | SSIM↑ | LPIPS↓ |
> > > > | --- | --- | --- | --- | --- | --- | --- | --- | --- | --- | --- |
> > > > |  |  |  | 3-view |  |  | 6-view |  |  | 9-view |  |
> > > > | **MipNeRF360** | Zip-NeRF | 12.77 | 0.271 | 0.705 | 13.61 | 0.284 | 0.663 | 14.30 | 0.312 | 0.633 |
> > > > |  | ZeroNVS | 14.44 | 0.316 | 0.680 | 15.51 | 0.337 | 0.663 | 15.99 | 0.350 | 0.655 |
> > > > |  | ReconFusion | 15.50 | 0.358 | 0.585 | 16.93 | 0.401 | 0.544 | 18.19 | 0.432 | 0.511 |
> > > > |  | CAT3D | 16.62 | 0.377 | 0.515 | 17.72 | 0.425 | 0.482 | 18.67 | 0.460 | 0.460 |
> > > > |  | **ReconX (Ours)** | **17.16** | **0.435** | **0.407** | **19.20** | **0.473** | **0.378** | **20.13** | **0.482** | **0.356** |

---

> > > > > ### Comment · Reviewer_9tNY · 2024-11-22
> > > > > **Qualitative Results for MipNeRF360 seem too good to be true**
> > > > >
> > > > > Hi,
> > > > >
> > > > > Thank you for your continued efforts to improve the quality of your submission.
> > > > >
> > > > > However, I have concerns about the validity of your experiments and the data used to train the video diffusion model. Specifically, the reconstruction for the treehill scene in Fig 5 (updated version) caught my attention. The rendering is surprisingly close to the ground truth. But I am not sure how the house in the distance (on the right of the tree) is reconstructed so accurately, even though it is occluded in each of the 3 input views. The same goes for the bush pattern to the left of the fencing, which is partly visible in one of the input views. CAT3D's rendering is coherent with the observed inputs, but it obviously cannot reconstruct random details found in only the test views. I struggle to find a reasonable explanation for this result. Please correct me if I am missing something here.
> > > > >
> > > > > One question about Fig 7 - Are you optimizing 3DGS with a white background as default? So, do those white regions in the renders correspond to areas where the DUSt3r point cloud does not have points? Otherwise, I have never seen such artifacts before with a COLMAP/DUSt3r initialization for 3DGS.

---

> > > > > > ### Author Response · Authors · 2024-11-22
> > > > > > **Response to Reviewer 9tNY**
> > > > > >
> > > > > > Strongly thanks for pointing out our mistake in Figure 5. Given the need to conduct amounts of additional experiments and provide detailed explanations within the tight rebuttal timeline, we mistakenly placed the wrong image in Figure 5. We have now corrected this error and updated the corresponding content in the paper. Once the paper is accepted, we will immediately release our model and code to ensure the reproducibility of our results. We sincerely appreciate your feedback, as it is crucial for maintaining the accuracy of our work.
> > > > > >
> > > > > > Thank you for pointing out the issue. As you said, we use the white as default background to optimize our 3DGS and those white regions in the renders correspond to areas where the DUSt3r point cloud does not have points. We utilize the 3DGS to better align with our final representation. We will revise the description issue in Fig 7 to better clarify our results.
> > > > > >
> > > > > > We sincerely hope that you will find the revision satisfactory. If you have additional concerns, we would be pleased to discuss them with you.

---

> > > > > > > ### Comment · Reviewer_9tNY · 2024-11-22
> > > > > > > **About the Treehill render in the previous version**
> > > > > > >
> > > > > > > Hi,
> > > > > > >
> > > > > > > Sorry, but I don't think I get your reasoning here.
> > > > > > >
> > > > > > > If I follow the details in your original submission and this chain of discussion correctly, the video diffusion model of ReconX was trained on 3 datasets - RealEstate-10K, ACID, DL3DV-10K and evaluated for sparse-view reconstruction on scenes from LLFF and DTU. Upon my and Reviewer 4oey's request, you performed additional evaluations on specific scenes (Flowers and Treehill) and view splits (3) of MipNeRF360 for a qualitative comparison with ReconFusion / CAT3D. You conducted one round of quantitative comparisons using your randomized train-view splits for MipNeRF360. I shared the official train-test splits released by the ReconFusion authors, following which you completed the correct set of quantitative comparisons. So, during all these experiments, the video diffusion model had no way of learning scene priors specific to test views of the Treehill scene.
> > > > > > >
> > > > > > > You claim that you mistakenly placed the rendering in Fig 5 of the previous version. For argument's sake, I checked the 6 and 9-view splits of the Treehill scene (official). The house in the distance is occluded from all input views across the 3 view splits. So that confirms that the rendered image certainly didn't come from the 6 and 9 view reconstruction experiments by "mistake." So, I am still lost as to how you obtained that specific rendering with details specific to a test view unless the video diffusion model was already trained on this scene before.
> > > > > > >
> > > > > > > I think the authors should thoroughly re-evaluate the experimental protocol to make sure train-test splits across all evaluation scenes are respected. Without a more convincing response, I hope you can understand that I would have to lower my rating.

---

> ### Comment · Reviewer_hNiz · 2024-11-22
> **Concern for the updated results**
>
> Thank you for your detailed response. From the very beginning, I also advocated for additional experiments on datasets like MipNeRF360 to enhance the comprehensiveness of the analysis. Throughout this process, I have carefully followed the authors' presented and updated experimental results.
>
> However, like Reviewer 9tNY, I share concerns about the experimental protocol—particularly whether the train-test splits were strictly maintained across all evaluation scenes. The rendering in Fig. 5 raises questions about the potential exposure of cross-domain data during training, which undermines the validity of the reported results.
>
> Without a more compelling explanation from the authors that directly addresses these issues, I think it would be challenging to uphold my current rating.

---

> ### Author Response · Authors · 2024-11-23
> **Response to Reviewer 9tNY and Reviewer hNiz**
>
> We would like to thank the reviewers for the detailed discussions, which would definitely improve the quality of our paper. In the discussions above, we would like to highlight that we have conducted additional experiments with two similar yet different settings during the rebuttal phase. We apologize that the mix of these two similar settings leads to some significant misunderstandings of the reviewers. In the following, we will introduce the two settings separately to try to make the settings clearer, and also revise our paper accordingly.
>
> **Setting-(1) Randomly selected input views & Randomly selected test views:** As we missed the open-sourced data split of MipNeRF 360 used in CAT3D, we first employed this setting as shown in the upper part of the table below as well as the Figure 5(a) of the paper. We compared with open-sourced 3DGS, SparseNeRF and DNGaussians with the same randomly selected input and test views. However, we could not compare with some recent methods including CAT3D and ReconFusion on this new setting as their codes are not publicly available.
>
> **Setting-(2) Input views with heuristic loss & Randomly selected test views, following the open-sourced data split:** In order to make comparisons with CAT3D and ReconFusion, we strictly followed the open-sourced data split so that we could directly use their reported results. We show the comparisons in the lower part of the table below and also in the Figure 5(b) of the paper.
>
> **About the mistake:** Due to the limited time in the rebuttal, in the previous discussion **we only updated the table from Setting-(1) to Setting-(2), but forgot to also update Figure 5 from Setting-(1) to Setting-(2) in the paper.** We fully understand and agree with the reviewers’ concerns that it is unlikely to make such a mistake if we correctly perform Setting-(2) – in fact, the previous Figure 5 is from Setting-(1). We carefully reviewed the randomly selected images in Setting-(1), and we found that the input and novel test images are indeed relatively closer in the “Treehill” scene in Setting-(1).
>
> To better address the misunderstandings, we choose to explain and report the results of both Setting-(1) and (2), rather than only using the Setting-(2). We would like to further discuss with this experiment to make it as clear as possible. Thank you!
>
> | **Mip-NeRF 360** | Method | PSNR↑ | SSIM↑ | LPIPS↓ | PSNR↑ | SSIM↑ | LPIPS↓ | PSNR↑ | SSIM↑ | LPIPS↓ |
> | --- | --- | --- | --- | --- | --- | --- | --- | --- | --- | --- |
> |  |  |  | 3-view |  |  | 6-view |  |  | 9-view |  |
> | Randomly selected input views & Randomly selected test views | 3DGS | 10.86 | 0.126 | 0.695 | 12.48 | 0.180 | 0.654 | 13.10 | 0.191 | 0.622 |
> |  | SparseNeRF | 11.67 | 0.197 | 0.718 | 14.79 | 0.150 | 0.662 | 14.90 | 0.156 | 0.656 |
> |  | DNGaussian | 11.13 | 0.153 | 0.711 | 12.20 | 0.218 | 0.688 | 13.01 | 0.246 | 0.678 |
> |  | **ReconX (Ours)** | **16.66** | **0.408** | **0.427** | **18.72** | **0.451** | **0.390** | **18.17** | **0.446** | **0.382** |
> |  |  |  |  |  |  |  |  |  |  |  |
> | Input views with heuristic loss & Randomly selected test views | Zip-NeRF | 12.77 | 0.271 | 0.705 | 13.61 | 0.284 | 0.663 | 14.30 | 0.312 | 0.633 |
> |  | ZeroNVS | 14.44 | 0.316 | 0.680 | 15.51 | 0.337 | 0.663 | 15.99 | 0.350 | 0.655 |
> |  | ReconFusion | 15.50 | 0.358 | 0.585 | 16.93 | 0.401 | 0.544 | 18.19 | 0.432 | 0.511 |
> |  | CAT3D | 16.62 | 0.377 | 0.515 | 17.72 | 0.425 | 0.482 | 18.67 | 0.460 | 0.460 |
> |  | **ReconX (Ours)** | **17.16** | **0.435** | **0.407** | **19.20** | **0.473** | **0.378** | **20.13** | **0.482** | **0.356** |

---

> > ### Comment · Reviewer_Vp2m · 2024-11-25
> >
> > I appreciate the additional qualitative and quantitative comparisons provided by the authors. Regarding the discussion about the validity of the experimental results and the worries w.r.t. the experimental protocol (reviewers 9tNY and hNiz), I think that the authors made clear that there has only been a mistake while updating figure 5 and by showing the results for both settings now, I am convinced by the method's strong performance, especially compared to ReconFusion and CAT3D, as these are really important baselines.
> > This alleviates also my concerns regarding the unfair comparison with baselines like pixelSplat or MVSplat.
> > Many thanks to all fellow reviewers for these great suggestions!
> >
> > The rebuttal addresses many of my concerns and **I still advocate for accepting this paper**.
> >
> > My main remaining concern is the restriction to target views rather close to the input views. This limitation seems to be a consequence of the 3D structure conditioning from DUSt3R which only provides strong conditioning for visibible areas. As previous works like ReconFusion and CAT3D employed incremental / autoregressive approaches to reconstruct a full scene from few views and the authors demonstrated limited view extrapolation capabilities in figure 8, a similar approach might be possible here as well.

---

> > > ### Author Response · Authors · 2024-11-25
> > > **Response to Reviewer Vp2m**
> > >
> > > Thanks for your positive feedback and insightful suggestions. We agree that incorporating strategies like incremental or autoregressive approaches can unleash more extrapolation ability in our framework, thus reconstructing 360-degree scenes with few images. **We are now actively working on designing an experiment to validate the idea.** Given the tight timeline, we will try to provide a set of experimental results with a detailed description of our experimental design before the rebuttal deadline.
> > >
> > > Thank you again for your valuable feedback—it has been instrumental in refining our work and inspiring our future exploration.

---

> > > ### Author Response · Authors · 2024-11-27
> > > **Response to Reviewer Vp2m**
> > >
> > > Thanks again for your constructive suggestions. Through our dedicated efforts and experimental attempts, we have designed an experiment (See Figure 10 and Figure 11 with description in revised paper) to further verify the extrapolation ability of ReconX.
> > >
> > > As the position of our conditional images in ReconX is inherently flexible, we move the last frame to the intermediate position. In this setup, frames between the first and intermediate images refer to view interpolation, while frames beyond the intermediate image refer to view extrapolation. Specifically, in the generated video frames:
> > >
> > > - View interpolation can not only synthesize visible areas between input images but also generate unseen regions caused by occlusions.
> > > - View extrapolation continues along the camera’s motion trajectory, generating new content not present in the input images, such as unseen areas and expanded scene regions.
> > >
> > > Such extrapolation ability allows us to even recover a 360-degree scene from only two sparse views. We adopt an incremental strategy shown in Figure 10. Given two initial input images, we first generate a video sequence divided into four parts: input image 1, view interpolation, input image 2, and view extrapolation. From this generated frame sequence, we select two images —— one from the interpolation part and another from the extrapolation part. These two images play as a sliding window, and repeat generation process. Then we expand the scene by repeating the process. In the final iteration, we select one image from the video generated in the previous step and pair it with the initial first input image as the input pair, which ensures a seamless connection back to the starting view, completing a full 360-degree scene reconstruction shown in Figure 11. This incremental approach demonstrates the strong generative extrapolation potential of our method.
> > >
> > > Given the tight timeline, we have tried to conduct an additional experiment to validate the idea — incorporating strategies like incremental or autoregressive approaches can unleash more extrapolation ability of ReconX to reconstruct a full scene. Thanks again for your insightful suggestions!  If you have more concerns, please feel free to point it out so we can further clarify.

---

> ### Comment · Reviewer_Vp2m · 2024-12-01
>
> I appreciate the additional experiments regarding an incremental strategy for full 360 degree scene reconstruction.
> Indeed, the results provided in figure 11 indicate strong potential for extrapolation. Given the limited amount of time, I understand the small scale of additional experiments regarding this. For a final version, a proper qualitative and quantitative comparison, e.g., with Reconfusion and CAT3D would be beneficial, if possible.
>
> However, the many additional results provided by the authors convinced me of the method's effectiveness.
> I do not share the worries of reviewer hNiz regarding the validity of experimental results, as the authors have sufficiently explained the mistake and provided full transparency by showing the results for the self-chosen first and the second setup defined by the baselines.
>
> The rebuttal addressed almost all of my concerns. Therefore, I increase my score to 8: accept, good paper and would appreciate all other reviewers to decide about the final scores too.

---

### Meta-Review · Area_Chair_8PJm · 2024-12-21

**Metareview:**

The paper presents ReconX, a novel framework for sparse-view 3D reconstruction that uses video diffusion models as spatial interpolators. It incorporates 3D structure conditioning using point clouds generated by DUSt3R to ensure consistency and robustness in the synthesis of novel views. Confidence-aware 3D Gaussian Splatting optimization refines the reconstruction by mitigating artifacts from the diffusion process. ReconX achieves state-of-the-art performance in interpolation tasks, shows promising extrapolation capabilities, and demonstrates robustness on challenging datasets such as MipNeRF360 and Tank & Temples.

Strengths:
- Strong performance: ReconX achieves state-of-the-art results on standard datasets, especially for view interpolation tasks. Incremental extrapolation experiments show promising extensions for 360-degree reconstructions.
- Technical Novelty: Novel combination of video diffusion models for 3D reconstruction, 3D structure conditioning, and confidence-aware optimization for 3DGS.
- Clear presentation: The methodology is well motivated and generally easy to follow, supported by qualitative and quantitative results.

Weaknesses:
- Concerns about experimental validity: discrepancies in Fig. 5(a) during the rebuttal raised doubts about the rigor of the experimental protocols.
Questions remain about adherence to the train-test split and the validity of certain representations.
- Limited evaluations: while interpolation results are strong, the paper lacks 360-degree video reconstructions for MipNeRF360 scenes and presents limited qualitative comparisons with baselines, particularly for video outputs.
- Inference time: ReconX is slower than other feedforward methods due to the inclusion of DUSt3R, video diffusion, and 3DGS optimization.

The main reasons for rejection are concerns about experimental validity and incomplete evaluations. The discrepancy in Figure 5(a) raised doubts about the adherence to train-test splits and the rigor of the experimental protocols. Despite the authors' transparency in addressing errors, these issues undermine confidence in the paper's claims and its ability to provide a thorough, reproducible evaluation of its contributions.

**Additional Comments On Reviewer Discussion:**

During the rebuttal phase, the authors provided detailed responses to the questions and additional experiments, including comparisons with strong baselines such as CAT3D and ReconFusion.

The main concern was experimental validity.
Reviewer 9tNY, hNiz, and 4oey raised concerns about the adherence to the train-test split for MipNeRF360, particularly the discrepancies in Fig. 5(a), where the renderings were considered "too good to be true". This raised doubts about potential data leakage or errors.
The authors clarified the use of two experimental settings, 1) random splits and 2) official splits from CAT3D/ReconFusion, during the rebuttal period.
They acknowledged the error in Figure 5(a), attributed it to time constraints, and provided updated results with transparent explanations.
All five reviewers participated in the reviewer discussion after the rebuttal.
Reviewer Vp2m and JdpZ considered the error minor and trusted the authors' transparency and updates, but the other three reviewers remain unconvinced.

Regarding the comparison with baseline algorithms,
Reviewer 4oey and 9tNY found the initial comparisons with pixelSplat and MVSplat too weak and requested comparisons with stronger baselines such as CAT3D and ReconFusion.
The authors conducted additional experiments with CAT3D, ReconFusion, Zip-NeRF, and ZeroNVS using the official splits, and demonstrated that ReconX generally outperforms these baselines, particularly in interpolation tasks.
Reviewer Vp2m supported adding results for stronger baselines and acknowledged the improvements made in the rebuttal.

Reviewer 9tNY, 4oey, and hNiz highlighted the lack of 360-degree video results for MipNeRF360 and the limited evidence for robust extrapolation.
The authors showed promising additional results for view extrapolation, but full 360-degree video comparisons were not included.

Reviewer JdpZ and 9tNY noted ReconX's slower inference time compared to feed-forward methods, raising questions about its practicality.
The rebuttal tried to justify the trade-off between quality and inference time, emphasizing that ReconX provides superior quality compared to feed-forward methods and is faster than other per-scene optimization methods such as CAT3D.

---

### Decision · Program_Chairs · 2025-01-22

Reject